# Tunable two-dimensional interfacial coupling in molecular heterostructures

Beibei Xu[1,2], Himanshu Chakraborty[3,4], Vivek K. Yadav[3], Zhuolei Zhang[1,2], Michael L. Klein[2,3] & Shenqiang Ren[1,2]

Two-dimensional van der Waals heterostructures are of considerable interest for the next generation nanoelectronics because of their unique interlayer coupling and optoelectronic properties. Here, we report a modified Langmuir–Blodgett method to organize two-dimensional molecular charge transfer crystals into arbitrarily and vertically stacked heterostructures, consisting of bis(ethylenedithio)tetrathiafulvalene (BEDT–TTF)/$C_{60}$ and poly (3-dodecylthiophene-2,5-diyl) (P3DDT)/$C_{60}$ nanosheets. A strong and anisotropic interfacial coupling between the charge transfer pairs is demonstrated. The van der Waals heterostructures exhibit pressure dependent sensitivity with a high piezoresistance coefficient of $-4.4 \times 10^{-6}\,Pa^{-1}$, and conductance and capacitance tunable by external stimuli (ferroelectric field and magnetic field). Density functional theory calculations confirm charge transfer between the $n$-orbitals of the S atoms in BEDT–TTF of the BEDT–TTF/$C_{60}$ layer and the $\pi^{\star}$ orbitals of C atoms in $C_{60}$ of the P3DDT/$C_{60}$ layer contribute to the inter-complex CT. The two-dimensional molecular van der Waals heterostructures with tunable optical–electronic–magnetic coupling properties are promising for flexible electronic applications.

[1] Department of Mechanical Engineering, Temple University, Philadelphia, PA 19122, USA. [2] Temple Materials Institute, Temple University, Philadelphia, PA 19122, USA. [3] Department of Chemistry and Institute for Computational Molecular Science, Temple University, Philadelphia, PA 19122, USA. [4] Center for the Computational Design of Functional Layered Materials, Temple University, Philadelphia,, PA 19122, USA. Correspondence and requests for materials should be addressed to S.R. (email: shenqiang.ren@temple.edu)

Two-dimensional (2D) heterostructures based on weak interlayer van der Waals (vdW) interaction with the lack of superficial dangling bonds afford multiple degrees of freedom for the creation of new high-quality 2D heterojunctions and superlattices without the constrains of lattice parameters, enabling customized and tunable optical–electronic–magnetic properties[1–5]. Currently, 2D heterostructures are predominantly based on inorganic complexes, especially chalcogenides with strong vertical chemical bonds[1]. In this context, the urgent demand of flexible nanoelectronics and optoelectronics calls for a novel generation of organic heterostructures held by vdW forces for both vertical and horizontal orientation[6, 7]. Among the organic complexes, the superior optoelectronic properties and the lack of interlayer screening effect for the 2D donor and acceptor CT molecular crystals, as well as the charge density wave induced long-range vdW force in the polarized structure, is a promising candidate to enable 2D CT molecular heterostructures with tunable optical–electronic–magnetic coupling behavior at the atomic level[8–17].

The atomically regulated interface of 2D heterostructures affords an ideal platform to understand the heterointerfacial coupling behavior. However, the conventional 2D hetero-structures are composed of two layers of opposite carrier type with CT interaction[2]. Therefore, it is critical to investigate the coupling behavior between two pairs of different CT layers, which holds promise to show unique coupling behavior distinct from traditional inorganic counterparts due to the inter-conversion between singlet and triplet CT states under external stimuli combined with the internal hyperfine interaction and spin-orbit coupling of organic semiconductors. Moreover, the combination of the distinct properties from different layers is not only com-plementary but also will enhance the performance of the whole system[18].

To prepare 2D vdW heterostructures (vdWH), conventional chemical vapor deposition approach is widely used. However, the weak interlayer interaction between the monolayer 2D film and the substrate leads to the island growth rather than continuous monolayers[1]. Moreover, the choice of lattice-matched substrate is needed for the growth of 2D materials, which limits the scalable preparation[6, 19, 20]. Thus, a universal method for the preparation of organic 2D vdWHs is in urgent demand. On the other hand, the understanding and control of heterointerfacial coupling behavior for the organic vdWHs is of key importance towards the further design and applications of the nano-devices. Among all the approaches to control the interfacial state, extrinsic charge injection by ferroelectric field generated from flexible P(VDF-TrFE) polymer has drawn much attention. The large polarization of the crystallized ferroelectric polymer, P(VDF-TrFE) induces a net polarization of organic CT components with enhanced optoelectronic and even magnetoelectric performance[21–24].

Here, we report the large scale assembly of 2D CT hetero-structures with controlled orientation and unique physicochem-ical properties, consisting of vertically stacked poly(3-dodecylthiophene-2,5-diyl) (P3DDT) donor with fullerene ($C_{60}$) acceptor layer ($DTC_{60}$) and bis(ethylenedithio)tetrathiafulvalene (BEDT–TTF) donor with $C_{60}$ acceptor layer ($ETC_{60}$). The interfacial coupling can be tuned in over a large range by external fields (ferroelectric and magnetic) with strong enhancement of current and capacitance, due to the coupling across two CT pairs along the vertical and horizontal orientations.

## Results

**Growth and structure of the vdWHs.** As shown in Fig. 1a, sol-vent vapor evaporation assisted by a modified LB approach is applied to facilitate the assembly and sequential stacking of uniform ~ mm$^2$ scalable free-standing $ETC_{60}$ and $DTC_{60}$ nanosheets. The mixed water/dimethylformamide (DMF) solvent with large surface tension, good spreading ability, and the low vapor pressure of 1,2-dichlorobenzene (DCB) solvent for $ETC_{60}$ and $DTC_{60}$ are all crucial for the assembly and stacking of 2D heterostructures (Fig. 1b, Supplementary Figs. 1–3). According to Marangoni flow, when a droplet of solution is dropped onto the surface of a high-surface-energy phase, local surface tension gradients will generate around the boundary of the droplet, leading to the surficial flow of the solution towards the higher surface tension part[25, 26]. The tendency to form an organic film on the surface of water is determined by the spreading pressure $S$, which is the difference of the surface tensions along the three-phase contact line—the surface tensions $\sigma_1$ and $\sigma_2$ of the dropped solution and water, and the interfacial tension $\sigma_{1,2}$ of the two phases:

$$S = \sigma_1 - \sigma_2 - \sigma_{1,2} \qquad (1)$$

For 1,2-DCB on the surface of water, the spreading pressure is −4.25 dyne cm$^{-1}$[27]. Thus, wetting is not complete, and drops form. By contrast, as DMF is miscible in both water and 1,2-DCB, the mixed water/DMF solution enables the spreading of the 1,2-DCB solution into a film (Supplementary Fig. 1b). The optical microscopy (OM) image (Fig. 1c), elemental mapping of scanning electron microscopy (SEM) image (the inset of Fig. 1c), the plane-view and 3D atomic force microscopy (AFM) images confirm the formation of vdWHs with an average surface roughness of ~ 0.5 nm (Fig. 1d, e). The average thicknesses of the $ETC_{60}$ and $DTC_{60}$ nanosheets are ~ 8 mm and 23 nm, respectively. By controlling the volume and the concentration of organic solution, the morphology and thickness of the nanosheets can be modulated (Supplementary Figs. 4 and 5).

The crystallized 2D nature of such heterostructures is further confirmed by transmission electron microscopy and selected-area electron diffraction patterns (Fig. 1f and the inset). XRD patterns reveal that the molecular chains of BEDT-TTF and P3DDT are both in the in-plane orientation (Supplementary Fig. 6). The shift of the Raman peak corresponding to the symmetric ($A_g$) vibration of $C_{60}$ in the $ETC_{60}$-$DTC_{60}$ heterostructure nanosheets compared to that in single $ETC_{60}$ and $DTC_{60}$ nanosheets confirms the existence of coupling between the $ETC_{60}$ and $DTC_{60}$ nanosheets (Supplementary Fig. 7). Thus, a schematic for the vdWH comprise of $ETC_{60}$-$DTC_{60}$ can be figured out as shown in Fig. 1g, h and Supplementary Fig. 8. The BEDT-TTF and $C_{60}$ molecules are alternatively stacked along the horizontal orientation (x and y axis), forming the segregated stacking 2D $ETC_{60}$ nanosheets with the distance between each layer close to 6.46 Å[28]. The P3DDT layers self-organize through the π-π interactions[16]. The distinct structure of P3DDT and $C_{60}$ molecules lead to the segregated stacking of P3DDT and $C_{60}$ in the 2D $DTC_{60}$ nanosheets, where the closest distance between the backbone of P3DDT and $C_{60}$ molecules is 8.08 Å[16]. Each $ETC_{60}$ nanosheet crystalline unit contains six molecules of BEDT-TTF and four molecules of $C_{60}$, while that of $DTC_{60}$ contains two molecules of P3DDT and two molecules of $C_{60}$. The orthorhom-bic crystalline unit cell of $ETC_{60}$/$DTC_{60}$ has lattice parameters of $a = 26.71$ Å, $\alpha = 90°$, $b = 32.51$ Å, $\beta = 90°$, $c = 58.90$ Å, $\gamma = 90°$. The horizontal stacking of the molecular chains on water surface forms a stable structure, which can also be verified by the following anisotropic properties. The CT interaction mainly exists along the vertical orientation, while π–π stacking and interchain interaction dominate the horizontal orientation.

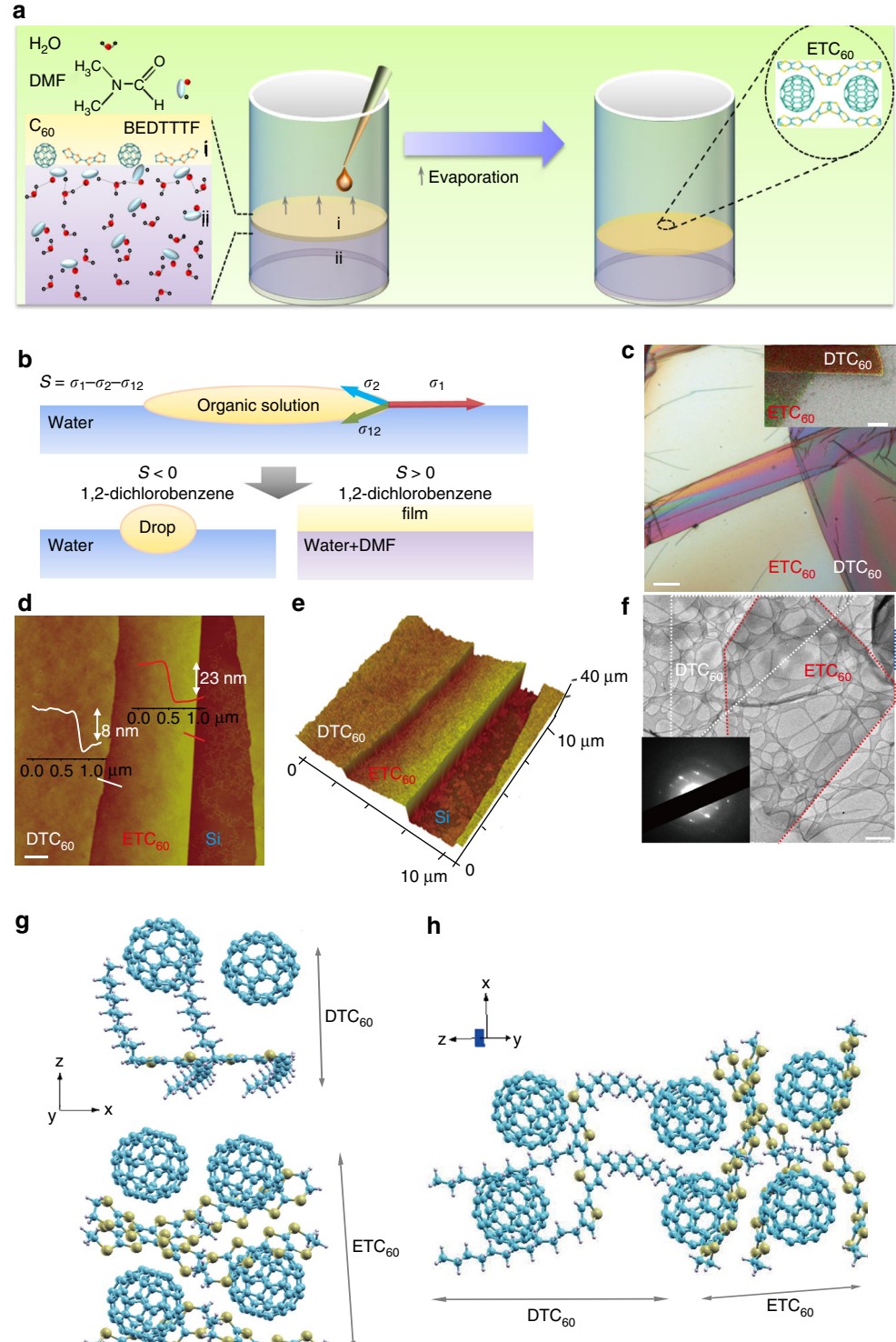

**Fig. 1** Growth, morphology, and structure of the vdW heterostructures. **a** Growth scheme of the BEDT–TTF/$C_{60}$ (ETC$_{60}$) nanosheet. BEDT–TTF/$C_{60}$ solution was drop-casted onto the surface of 50 vol%DMF/50%water mixed solution, with the diffusion and evaporatin of BEDT–TTF/$C_{60}$ solution, thin nanosheet forms. P3DDT/$C_{60}$ (DTC$_{60}$) nanosheet was grown in the same way. **b** Spreading pressure dependent film formation mechanism. **c** Optical microscopy (OM) image. The *light gray color* corresponds to ETC$_{60}$ nanosheet. **d**, **e** Plane-view and three dimensional view of atomic force microscopy (AFM) images. **f** Transmission electron microscopy (TEM) image and selected area electron diffraction (SAED) pattern. **g**, **h** Stacking structure of ETC$_{60}$ nanosheets and DTC$_{60}$ nanosheets along different axis. Molecular chains for BEDT–TTF and P3DDT are along the horizontal orientation. The dipoles and charge transfer are mainly along the vertical orientation. $z$ axis is along the out-of-plane orientation. $x$ and $y$ axes are along the in-plane orientation

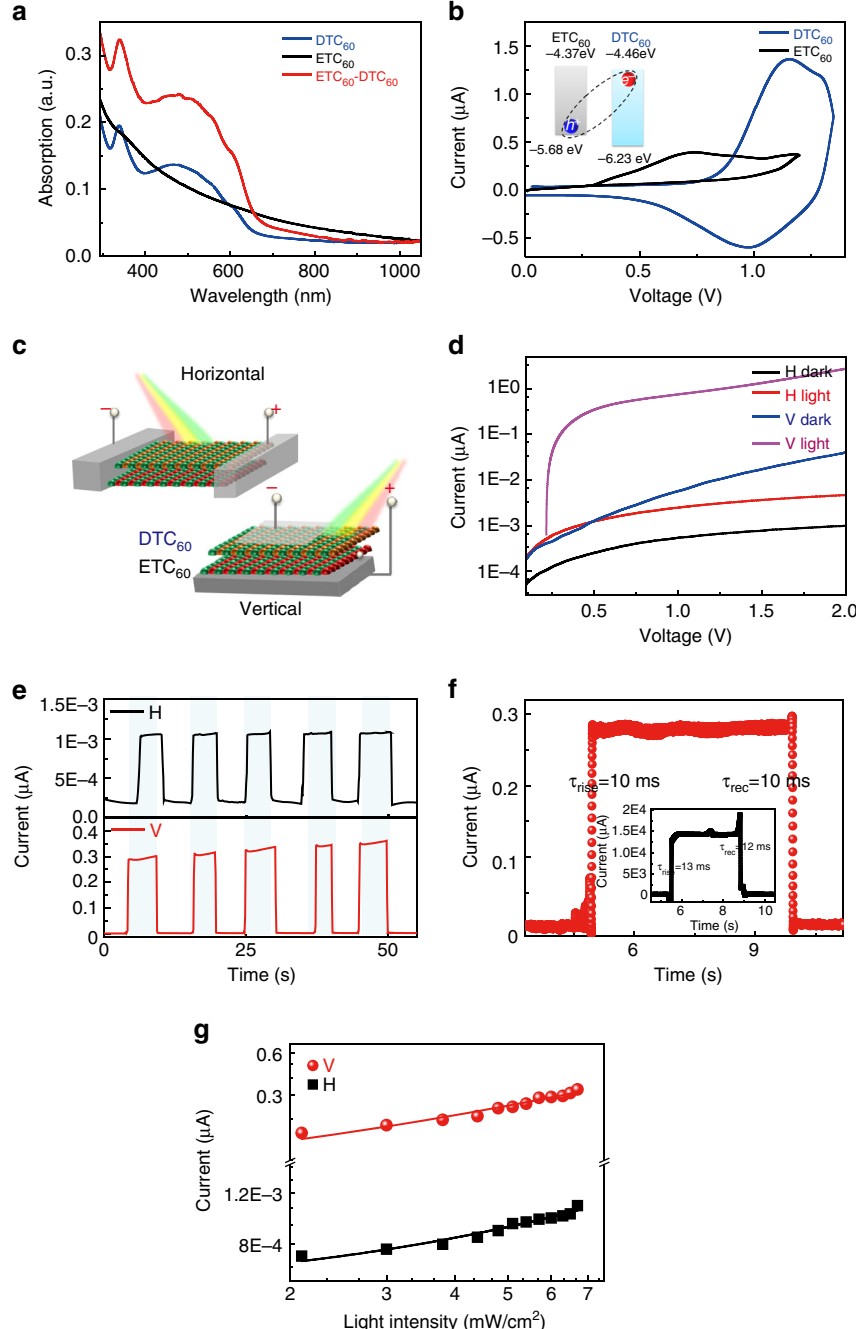

**Fig. 2** Anisotropic optoelectronic properties of the vdW heterostructures. **a** Absorption spectra of $ETC_{60}$, $DTC_{60}$ and the $ETC_{60}$-$DTC_{60}$ nanosheets.
**b** Cyclic voltammetry curves of $ETC_{60}$ and $DTC_{60}$ nanosheets, respectively. The inset shows the energy band diagram for HOMO/LUMO levels and the charge transfer interaction of $ETC_{60}$ and $DTC_{60}$ nanosheets. **c** Measurement scheme for the horizontal and vertical orientations. **d** Dark and light current-voltage curves under 365 nm light of 6.5 mW/cm². **e** Photoresponse with light on and off. **f** Photoresponse rise and fall rate for the vertical orientation. The inset is that for the horizontal orientation. **g** Light intensity dependent photocurrent. The *red lines* are the linearly fitted curves

**Optoelectronic properties**. The anisotropic stacking has a large influence on the optoelectronic properties of 2D vdW $ETC_{60}$-$DTC_{60}$ heterostructure devices. The optical, band alignment and anisotropic photoresponse properties are shown in Fig. 2. The highest occupied molecular orbital (HOMO) level of both nanosheets can be obtained by the cyclic voltammetry curves (Fig. 2b). In combination with the bandgap obtained from optical spectra (Fig. 2a), the HOMO and lowest unoccupied molecular orbital (LUMO) of $ETC_{60}$ can be attributed to −5.68 and −4.37 eV, respectively. In addition, the HOMO and LUMO levels of the $DTC_{60}$ nanosheet are −6.23 and −4.46 eV, respectively. The

energy offset between $ETC_{60}$ and $DTC_{60}$ nanosheets facilitates the CT interaction for the coupling between $ETC_{60}$ and $DTC_{60}$ nanosheets (the inset of Fig. 2b). The broad-spectrum absorption of $DTC_{60}$ nanosheets results in the broadband photoresponse of the vdWH $ETC_{60}$-$DTC_{60}$ (Supplementary Figs. 14 and 15). Figure 2c shows the measurement scheme for optoelectronic properties along the horizontal and vertical orientations. For both orientations, charge transport through all the molecules, and thus they reflect the average effect of the combination of all molecules. The influence of charge trap effect of the solvent can be excluded (Supplementary Fig. 10). Current–voltage curves along the

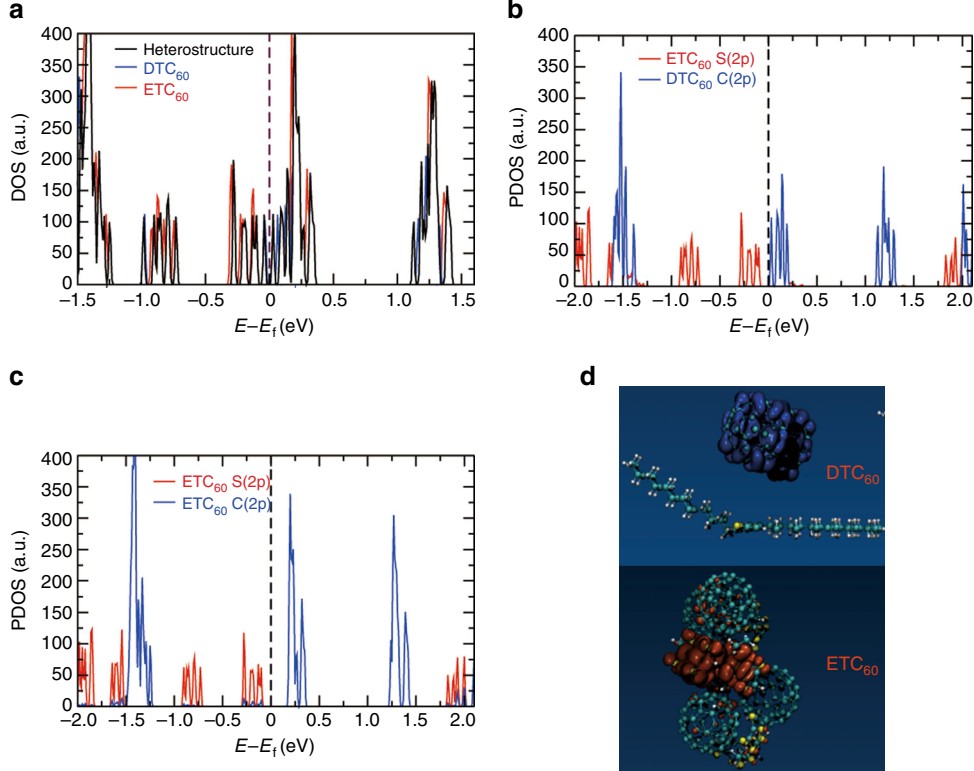

**Fig. 3** First principle DFT calculation on the nature of CT state. **a** Electronic density of states (DOS) plots. Solid curve denotes the DOS of $ETC_{60}/DTC_{60}$ heterostructure (*black curve*), *red curve* for $ETC_{60}$ complex and *blue curve* for $DTC_{60}$ complex. The energies have been shifted with respect to their Fermi energies. **b, c** Projected density of states (PDOS) plots. *Solid curves* denotes the PDOS of $ETC_{60}/DTC_{60}$ heterostructure for S (*red*). In Fig. 3b, the *blue curve* is for C from $C_{60}$ of the $DTC_{60}$ complex respectively. In Fig. 3c, the *blue curve* is for C of $ETC_{60}$ complex. The energies have been shifted with respect to their Fermi energies. **d** Charge density isosurface of the HOMO (*red*) and LUMO (*blue*) bands of the $ETC_{60}/DTC_{60}$ heterostructure

horizontal and vertical orientation under dark and 365 nm illumination are shown in Fig. 2d. In consideration of the dimensions, the calculated resistivity of the heterostructure along the horizontal and vertical orientations is $9.2 \times 10^7$ and $2.5 \times 10^{11}$ Ω cm, respectively, which are over 100 times lower than those of $ETC_{60}$ single nanosheet (Supplementary Fig. 13), demonstrating the enhancement effect of charge transport by the coupling behavior between $ETC_{60}$ and $DTC_{60}$ nanosheets. The rich π–π stacking and interchain interaction along the horizontal orientation provides the chain in the horizontal orientation with large density of charge, facilitating the transport of charge carriers. In contrast, the relatively larger distance between molecules and the larger discrepancy of molecular structures with relatively smaller overlap between different layers results in the lower charge density and charge transport along the vertical orientation, leading to a much larger resistance[16, 29–31]. The conducting AFM images for both horizontal and vertical orientations on the same device confirm the anisotropic conductivity (Supplementary Fig. 11). According to the photoresponsivity formula $\Delta J/P$, where $\Delta J$ is the current density difference between the light and dark conditions, $P$ is the power density of illumination, the horizontal and vertical orientation demonstrate a moderate photoresponsivity of 1.2 and 4.5 mA/W at 1.6 kV/cm, respectively (Fig. 2e). The matched band energy and the dominant CT along the vertical orientation make this orientation with a larger photoresponsivity. The rising and falling time of the photoresponse for the vertical orientation are both 10 ms, while those of the horizontal orientation are 13 and 12 ms, respectively (Fig. 2f). The external quantum efficiency (EQE) can be calculated by the formula $hcR/(e\lambda)$, where $h$ is planck's constant, $c$ demonstrates the speed of light, $R$ is photoresponsivity, $e$ represents the charge of an electron, $\lambda$ refers to the wavelength of the irradiated light. The EQE for the horizontal and vertical orientations are 41% and 155%, respectively. According to the photocurrent formula

$$I = AP^\theta, \tag{2}$$

where $\theta$ is the fitted slope related to the trapping and recombination process for the photogenerated charge carriers[32]. The $\theta$ values for the horizontal and vertical orientations are 0.961 and 0.959 under light intensity lower than 7 mW/cm², which is close to one unit, revealing that monomolecular recombination is the dominant recombination process at low light intensity (Fig. 2g)[32, 33]. Under a larger light intensity illumination up to 100 mW/cm², the relationship is not linear due to the loss of charge carriers via bimolecular recombination and space charge limited photocurrent from the unbalanced transport of electrons and holes[34, 35], which also corresponds to the potential light soaking effect (Supplementary Fig. 14e, and Supplementary Figs. 19 and 20). Moreover, the photoresponse can also be enhanced by external magnetic field and depends strongly on the stacking sequence of the different layers (Supplementary Figs. 14–18).

**Density functional theory calculations on the nature of CT state.** To understand the nature of CT in the heterostructure ($ETC_{60}$-$DTC_{60}$), we performed first principles density functional theory (DFT) calculations. The electronic density of states (DOS) and the partial density of states (PDOS) of the $ETC_{60}$-$DTC_{60}$ are shown in Fig. 3. The energies are shifted by the respective Fermi energy (Fig. 3a–c). The formation of the $ETC_{60}$-$DTC_{60}$ heterostructure results in shifting of the LUMO level closer to the Fermi energy. This decreases the HOMO-LUMO gap, and thus,

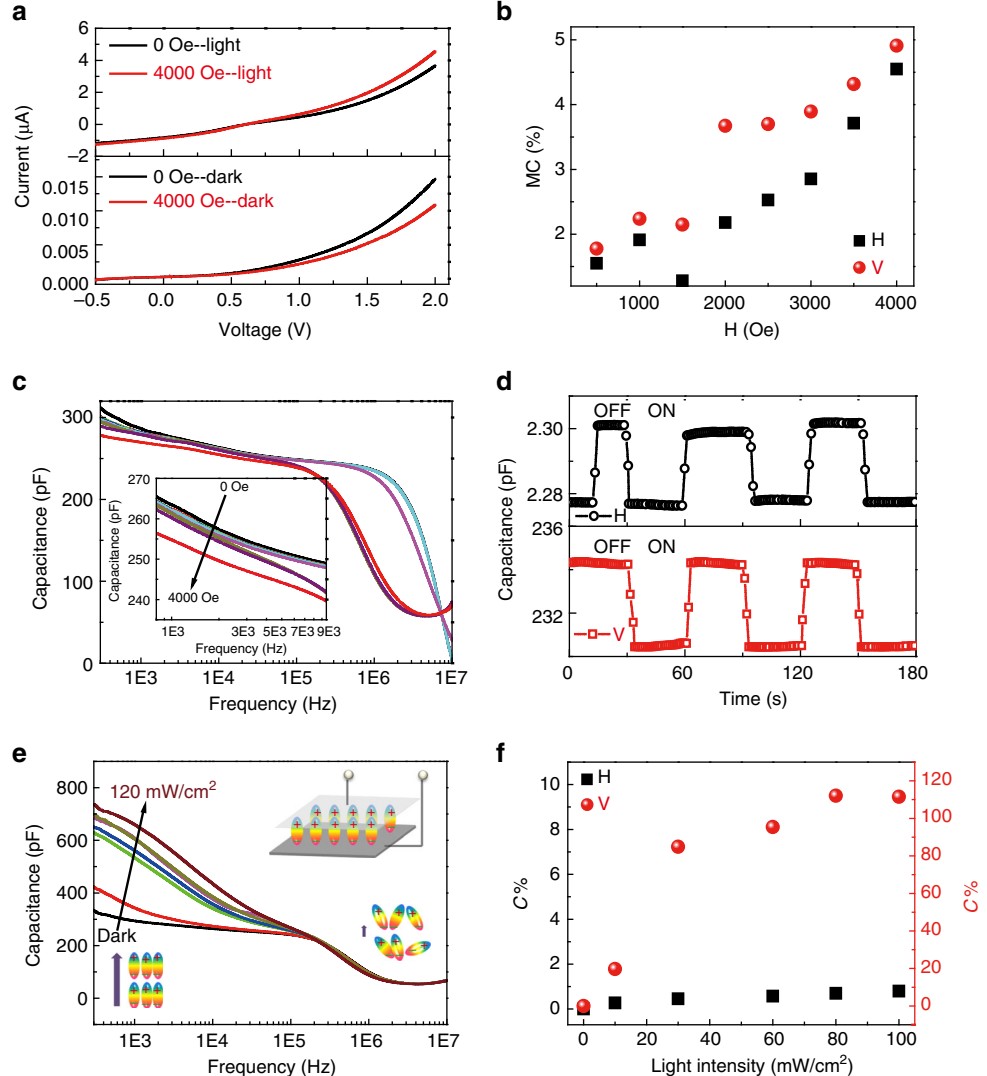

**Fig. 4** Magnetic field effect and light excited capacitance response. **a** Current–voltage curves for the vertical orientation under dark and light without and with the loading of a magnetic field of 4000 Oe. **b** Magnetic field dependent MC change for both orientations under dark. **c** Frequency dependent capacitance change under different magnetic field for the vertical orientation. **d** Capacitance change with magnetic field on and off for both orientations. The magnetic field is 4000 Oe. The frequency is 30 kHz. **e** Frequency dependent capacitance change under different light intensity for the vertical orientation. The inset is the dipole alignment along the electric field for the device and the dipole change as a function of frequency. **f** Light intensity dependent capacitance change for both orientations

contributes to the CT in the system. The HOMO is comprised mainly of the 2p states of S atoms in BEDT-TTF molecules of $ETC_{60}$, while the 2p states of C atoms in $C_{60}$ molecules of $DTC_{60}$ dominate the LUMO band (Fig. 3b). This suggests that CT occurs between the n-orbitals of the S atoms in BEDT-TTF of $ETC_{60}$ and the $\pi^*$ orbitals of C atoms in $C_{60}$ of $DTC_{60}$. Thus, the CT state is responsible for inter-complex charge transfer (between the $ETC_{60}$ and $DTC_{60}$ complexes) in the heterostructure. There is also a significant contribution of the intra-complex charge transfer within the $ETC_{60}$, but with a higher HOMO-LUMO gap than the inter-complex CT state (Fig. 3c). Other CT pathways of lower probability may also exist (Supplementary Figs. 21 and 22). The PDOS shows significant density of BEDT-TTF C atom 2p states in the HOMO, supporting the possibility of a $\pi$ to $\pi^*$ transition between BEDT-TTF and $C_{60}$ of $ETC_{60}$ and $DTC_{60}$, respectively. Intermolecular CT in this manner results in spatial separation of charge in the CT state, with accumulation of holes on the BEDT-TTF molecules and electrons on the $C_{60}$ molecules

of $DTC_{60}$. This is evidenced by the charge density isosurfaces of the HOMO and LUMO states in Fig. 3d.

**External stimuli dependent electrical properties change**. The anisotropic stacking can also influence the CT related physical properties. Among them, external field effects, including magnetic field effect and ferroelectric field effect, is an efficient and direct approach to reveal the CT process and interfacial coupling mechanism. Magnetic field effect can influence the derivation of current (MC)[36]. In addition, the transfer of electrons with the formation of dipoles and the anisotropic long-range ordered structure with the alignment of dipoles is able to influence the dielectric properties of the vdW heterostructures. Figure 4a shows the current-voltage curves along the vertical orientation under dark and light illumination with and without magnetic field of 4000 Oe. Under dark, the current decreases with the application of magnetic field, while it increases under light illumination. The current–voltage curves along the horizontal orientation show the

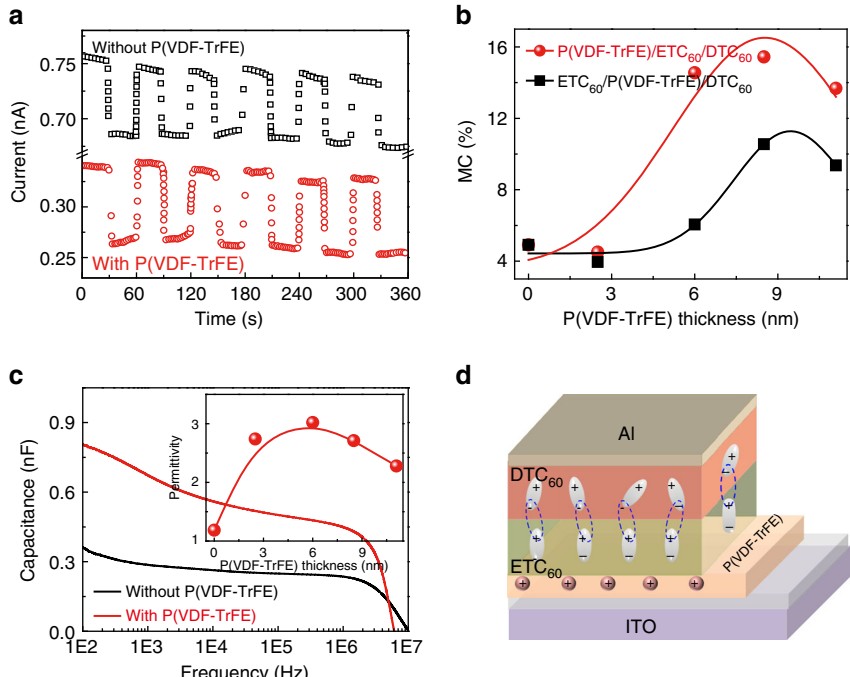

**Fig. 5** Vertical oriented electrical behavior under ferroelectric field. **a** Current change with magnetic field on and off for the heterostructure without and with the insertion of 8.5 nm P(VDF-TrFE) ferroelectric layer. The magnetic field is 4000 Oe. **b** P(VDF-TrFE) layer thickness dependent MC with P(VDF-TrFE) layer on the bottom of $ETC_{60}$ nanosheets ($PVDF/ETC_{60}/DTC_{60}$) and between $ETC_{60}$ nanosheets and $DTC_{60}$ nanosheets ($ETC_{60}/PVDF/DTC_{60}$), respectively. **c** Frequency dependent capacitance change without P(VDF-TrFE) layer, and with P(VDF-TrFE) layer on the bottom of $ETC_{60}$ nanosheets. The inset is the thickness of P(VDF-TrFE) layer dependent dielectric constant with P(VDF-TrFE) layer on the bottom of $ETC_{60}$ nanosheets. **d** Charge transfer, dipole alignment and the coupling between $ETC_{60}$ and $DTC_{60}$ nanosheets by the ferroelectric field effect of P(VDF-TrFE) layer

same tendency (Supplementary Fig. 23). MC can be calculated using the following equation[36],

$$MC = [I(H) - I(0)]/I(H), \quad (3)$$

where $I(H)$ and $I(0)$ is the current with and without magnetic field. The MC along vertical orientation is larger than that along horizontal orientation, and increases with the increase of magnetic field under dark (Fig. 4b). It also shows voltage and magnetic field dependent tendencies (Supplementary Fig. 24, Supplementary Figs. 26 and 27, and Supplementary Fig. 29). Magnetic field can induce the intersystem crossing from singlet CT to triplet CT, and the subsequent partial transformation to triplet excitons and polarons[37]. Under dark conditions, the scattering interaction between polaron and triplet exciton can decrease the mobility of polarons with the decrease of current density. By contrast, photoexcitation can increase the density of CT, leading to the increase of the scattering interaction and the dissociation of triplet excitons into free charge carriers with the increase of current density. Thus, the larger density of CT along the vertical orientation will ultimately cause larger MC than that of the horizontal orientation. The appliction of magnetic field will influence triplet excitons. As the lifetime of triplet exciton (~ μs) is much longer than that of singlet exciton (~ ps)[38], triplet excitons contribute to the formation of dipoles, and thus the capacitance can be changed by magnetic field, i.e., magneto-capacitance effect (Fig. 4c). Consequently, the large density of CT along the vertical orientation could inevitably increase the magneto-capacitance effect (Fig. 4d). The amplitude of the magneto-capacitance effect is comparable to inorganic single-phase multiferroic materials[39]. Fig. 4e presents the photoexcited capacitance change for the vdWHs along the vertical orientation. In the long-range ordered crystallized nanostructures, charge

transfer across the interface induces the generation of dipoles (triplet exciton). The measurement of capacitance can quantify the extent of the alignment of dipole moments (the inset of Fig. 4e). At a low frequency, the alignment of macroscopic dipoles (the charge polarization) can follow the ac electric field. Thus, the increased density of triplet excitons (dipole) under light illumination can lead to the increase of capacitance. At high frequency, although light could induce the increase of CT, the dipoles cannot follow the ac electric field, and as a consequence, the dipoles remain randomly oriented and do not contribute to the change of capacitance. Thus, the photoexcited capacitance change is calculated at a low frequency of 1000 Hz according to the formula[36]

$$C\% = [C(L) - C(0)]/C(0), \quad (4)$$

where $C(L)$ and $C(0)$ are the capacitance under light illumination and dark, respectively (Fig. 4f). Photoexcited capacitance shows overwhelming advantage for the vertical orientation with the maximum $C\%$ of 112% under the light intensity of 100 mW/cm$^2$. In comparison, $C\%$ for the horizontal orientation is below 2% due to the relatively lower density of dipoles along this orientation. Thus, the following investigations are based on the vertical orientation. By inversing the stacking sequence of the $ETC_{60}$ and $DTC_{60}$ nanosheets (laying the $DTC_{60}$ nanosheet close to ITO electrode and $ETC_{60}$ nanosheet close to Al electrode), the heterostructure can also demonstrate magneto- and photoexcited capacitance. However, as discussed in the inset of Fig. 2b, this kind of stacking is against the type-II band alignment, the heterointerface coupling between the CT of $DTC_{60}$ nanosheets and $ETC_{60}$ nanosheets is much weaker than that of the opposite stacking. Consequently, $C\%$ for magneto-capacitance and photoexcitation is lower than 1% and 4%, respectively (Supplementary Fig. 32).

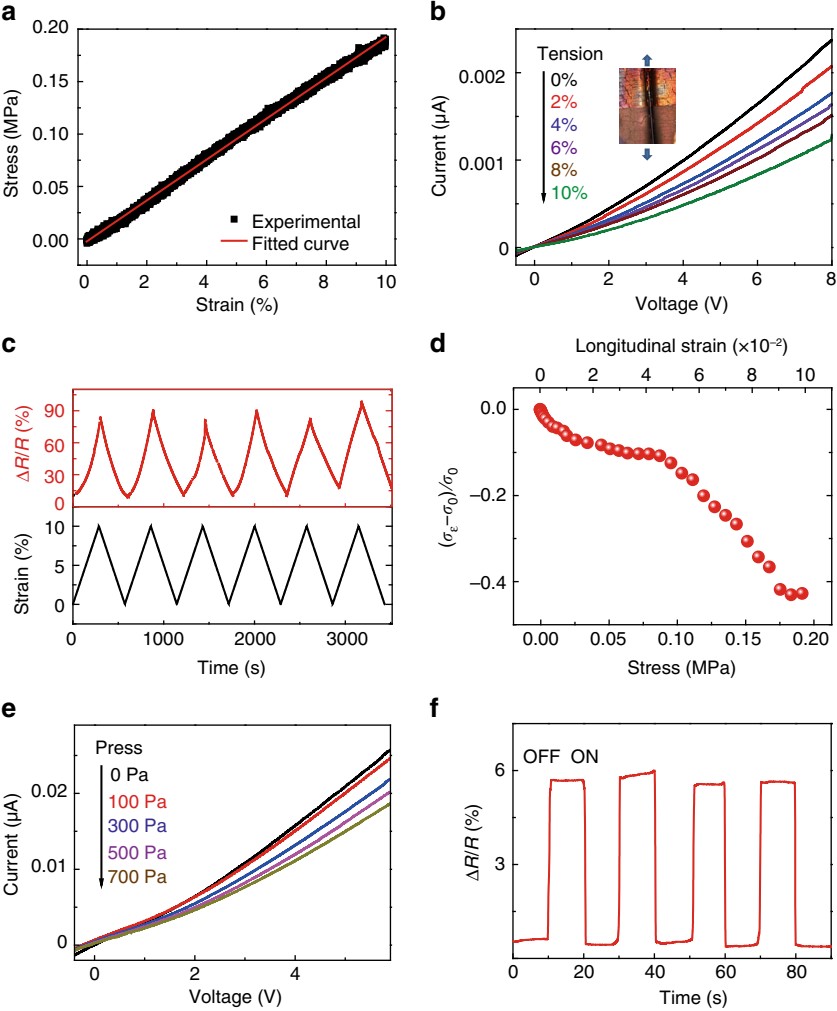

**Fig. 6** Horizontal oriented mechanical–electronical response. **a** Tension stress–strain relationship. The linear *red line* is the fitted curve. The inset is the digital photo image of the heterostructure on flexible PDMS substrate. **b** Tension strain dependent current–voltage curves with the strain from 0 to 10%. The inset is the OM image of the heterostructure on silver electrode with PDMS substrate. The tension direction is parallel to the electrode gap. **c** The periodical change of resistance corresponding to the 8 cycles of strain change. **d** Stress-strain dependent conductivity change. **e** Current–voltage curves with the loading of pressure from 0 to 700 Pa. **f** Resistance change with the cyclic on and off of 100 Pa pressure

To further understand the heterointerface coupling mechanism, the polarized P(VDF-TrFE) ferroelectric polymer with high electrostatic field are applied to tune the interfacial coupling. The ferroelectric field from the P(VDF-TrFE) layer enhances the CT density and MC effect (Fig. 5a, b). The MC of such heterostructures increases from 4.9% without P(VDF-TrFE) to an optimal value of 15.4% with P(VDF-TrFE) thickness of 8.5 nm underneath $ETC_{60}$ and $DTC_{60}$ nanosheets, confirming the tuning ability of the CT states by ferroelectric field. Two different device structures are compared to reveal the mechanism of interface coupling. For the first structure, the stacking sequence from bottom to top is P(VDF-TrFE) layer, $ETC_{60}$ nanosheet and $DTC_{60}$ nanosheet, respectively (Fig. 5d). The sequence of the other structure from bottom to top is $ETC_{60}$ nanosheet, P(VDF-TrFE) layer and $DTC_{60}$ nanosheet, respectively (Supplementary Fig. 35d). The stacking geometry plays an important role for the efficient MC, where the thin P(VDF-TrFE) layer inserted between $ETC_{60}$ and $DTC_{60}$ nanosheets shows a lower MC of 10.5% than that of P(VDF-TrFE) layer on the bottom of $ETC_{60}$ nanosheet due to the screening of CT interaction between $ETC_{60}$ and $DTC_{60}$ nanosheets by the inserted P(VDF-TrFE) layer (Fig. 5b), revealing

the importance of the interface between $ETC_{60}$ and $DTC_{60}$ nanosheets on the CT related properties. As discussed above, triplet exciton contributes to the formation of dipoles, thus the tuning of CT by P(VDF-TrFE) ferroelectric field influences the behavior of dipoles, leading to the change of dielectric behavior. As shown in Fig. 5c, the dielectric constant for the heterostructures with polarized P(VDF-TrFE) is much higher than that of heterostructures without P(VDF-TrFE). Dielectric constant decreases with the increase of frequency due to dielectric relaxation. The highest dielectric constant is obtained at 6 nm thickness of P(VDF-TrFE). When a ferroelectric P(VDF-TrFE) layer is inserted between $ETC_{60}$ and $DTC_{60}$ nanosheets, it also shows the same enhancement behavior (Supplementary Fig. 35d). As shown in Fig. 5d, the ferroelectric field effect from the P(VDF-TrFE) layer can enhance the CT and the alignment of dipoles in $ETC_{60}$ nanosheets, which then improves the interfacial coupling of dipoles between the $ETC_{60}$ and $DTC_{60}$ layers with enhanced MC and dielectric behavior.

**Mechanical-electrical response**. The outstanding flexibility and compatibility with scalable processing makes organic materials,

especially organic 2D polymers, affordable for flexible nanoelectronic applications[6, 40]. Two approaches are proposed to demonstrate the flexible electronic behavior of such 2D CT vdWHs, including tension strain and compress strain along the horizontal orientation. The inset of Fig. 6a shows the OM image for the prepared flexible device, where the tension stress is loaded parallel to the electrode gap. The tension stress–strain follows a repeatable linear relationship within the maximal strain of 10% (Fig. 6a), indicating the large flexibility of the heterostructure. The current decreases gradually with the increase of tensile strain (Fig. 6b) due to the increased interchain distance and decreased charge density with extended π–π stacking distance. The resistance changes periodically with the cyclic tension strain change between 0 and 10% (Fig. 6c), revealing the reversible behavior of the device. The enhancement of the resistance reaches the largest value of 85%. The strain sensitivity can be evaluated by the gauge factor $\frac{R_\varepsilon - R_0}{R_0 \varepsilon}$, where $R_\varepsilon$ and $R_0$ is the resistance under strain and without strain, $\varepsilon$ is strain. The maximal gauge factor is 8.5. The flexibility can also be expressed by the piezoresistance coefficient[41] according to

$$\pi^\sigma = \frac{1}{X} \frac{\sigma_\varepsilon - \sigma_0}{\sigma_0} \qquad (5)$$

where $X$ is the stress, $\sigma_\varepsilon$ and $\sigma_0$ is the conductivity under strain and without strain. The piezoresistance coefficient is $-4.4 \times 10^{-6}$ Pa$^{-1}$ below a strain of 4.3%, and it decreases to $-4.1 \times 10^{-6}$ Pa$^{-1}$ under higher strain. The piezoresistance coefficient of such 2D heterostructures is much larger than that of one dimensional nanowires[42]. The repeatability of the mechanical flexibility is confirmed by the pressure dependent electronic behavior (Fig. 6e). The pressure is loaded along the vertical orientation of the heterostructure, which also increases the interchain distance and decreases the charge density with extended π–π stacking distance, leading to the increase of resistance. Under a cyclic 100 Pa of pressure, the resistance increases immediately with the largest change of 5.7% (Fig. 6f).

## Discussion

In conclusion, solvent vapor evaporation assisted by the modified LB method is applied to organize two pairs of charge transfer layers, composed of poly(3-dodecylthiophene-2,5-diyl) (P3DDT) donor with a fullerene acceptor layer and bis(ethylenedithio)tetrathiafulvalene donor with a fullerene acceptor layer into scalable free-standing sequential stacking vdWHs. The coupling between two CT layers endows the heterostructure with broadband fast photoresponse of milliseconds. The anisotropic packing structure leads to a much larger coupling extent of charge transfer along the vertical orientation than that of horizontal orientation, leading to anisotropic optical–electronic–magnetic stimulated optoelectronic properties. Moreover, the ferroelectric field effect from P(VDF-TrFE) is applied to tune the coupling and charge transfer in the heterostructures. The DFT calculations confirm the charge transfer between the n-orbitals of the S atoms in BEDT-TTF of ETC$_{60}$ layer and the π* orbitals of C atoms in C$_{60}$ of DTC$_{60}$ layer contributes to the inter-complex charge transfer. The optimized heterostructure demonstrates high piezoresistance coefficient of $-4.4 \times 10^{-6}$ Pa$^{-1}$ under a tensile strain of 4.3%, and resistance increase by 5.7% under a small pressure of 100 Pa. The concept of heterointerface coupling between organic CT pairs provides a new 2D platform for fundamental research of the external stimuli dependent interaction of charges and excitons at atomic scale and the next generation of optoelectronic devices with unique functionality.

## Methods

**The preparation of BEDT-TTF/C$_{60}$ and P3DDT/C$_{60}$ solution.** 5 mg/mL BEDT-TTF (Tokyo Chemical Industry Co., LTD.) was dissolved in 1,2-dichlorobenzene (1,2-DCB) solvent, and stirred at 80 °C for 3 h. Then, 5 mg/mL C$_{60}$ (Sigma-Aldrich Co.) was added to above solution and stirred at room temperature for 2 h. After centrifuged at 6000 rpm for 20 min, the supernatant solution was isolated and used for the preparation of the film. For P3DDT/C$_{60}$ solution preparation, 5 mg/mL P3DDT (Ossila Ltd.) was dissolved in 1,2-DCB at 70 °C for 3 h. Then, 5 mg/mL C$_{60}$ was added into the solution and stirred at room temperature overnight.

**The preparation of 2D nanosheets.** In total 50 μL BEDT-TTF/C$_{60}$ solution was dropped slightly onto the surface of 20 mL distilled water/DMF mixed solution (volume ratio of 1:1) in a glass vessel and dispersed homogeneously. After several hours of slowly evaporation of 1,2-DCB solvent, thin two-dimensional nanosheet forms. The preparation method for P3DDT/C$_{60}$ nanosheet is the same, except that only 20 μL solution is needed. The preparation details for the heterostructure devices, the preparation of P(VDF-TrFE) layer and the flexible devices on PDMS substrates for mechanical characterization are discussed in Supplementary information.

**Morphology characterization.** Topography images were taken by Bruker dimension icon atomic force microscopy (AFM) under tapping mode with Si probe at a scanning rate of 1μm/s. The radius and resonant frequency are 25 nm and 13 kHz, respectively.

**Electrical properties characterization.** CHI 422 Series Electrochemical Workstation was used to get current and voltage signals. Driel's 50–150 W Research Arc Lamp Sources was used to output simulated solar light. In addition 365 nm, 650 nm, and 850 nm light comes from UVL-21 compact UV lamp, THORLABS red light and THORLABS LIU 850 A near-infrared light, respectively. The signal was obtained at a sample interval of 0.001 s.

**Cyclic voltammetry measurement.** Cyclic voltammetry experiment was carried out in a three-electrode cell with glassy carbon electrode as the working electrode, Pt electrode as counter electrode, and Ag/AgCl electrode in saturated KCl water solution as the reference electrode. All the electrodes were cleaned and blown dry by nitrogen gas before use. The 0.1 M tetrabutylammonium tetrafluoroborate (Sigma Aldrich) was used as the electrolyte and dissolved in dehydrated acetonitrile (ACN). The stability of Ag/AgCl electrode was checked by ferrocene (Sigma Aldrich) as internal standard substance. The ETC$_{60}$ and DTC$_{60}$ solution were self-assembled onto the working electrode and dried at 40 °C in glove box to form uniformly coated nanosheets. The measurement step was 25 mV/s. The HOMO level was obtained from the equation: $E_{HOMO} = [-\exp(E_{onset}$ (vs.Ag/AgCl)$-E_{onset}$ (F$_c$/F$_c$ + vs.Ag/AgCl))]$-4.8$ eV. The LUMO level was calculated from the obtained bandgap from the optical spectra.

**Capacitance and dielectric measurements.** Capacitance signal from 40 Hz to 10 M Hz was captured by Agilent 4294a Precision impedance equipment at 0.2 V and room temperature. Samples were collected by 16047E fixture. P(VDF-TrFE) ferroelectric layer was polarized by Keithley 2400 source meter at 15 V for 10 s.

**Mechanical flexibility characterization.** Tension stress was loaded by Instron 5944 Single Column Tabletop Low-Force Universal Testing System at a speed of 0.5 μm s$^{-1}$. Pressure was loaded by finger print. The pressure through finger was weighed and calibrated by a balance.

**Computational details.** Our calculations are based on first-principles density functional theory (DFT) as implemented in the Quantum ESPRESSO code[43]. We use a generalized gradient approximation (GGA) with the Perdew–Burke–Ernzerhof (PBE)[44] parameterization of exchange correlation energy functional with ultrasoft pseudopotentials[45]. An energy cutoff of 30 Ry was used for truncating the plane wave basis set to represent wave functions. Our recent work on ETC$_{60}$ crystals gives excellent result with these parameters[46]. The structures were relaxed until the magnitude of the Hellman–Feynman force on each ion became smaller than 0.03 eV/Å. Brillouin zone integrations were carried out with a uniform small mesh of k-points. The Grimme parameterization[47] was used to include the vdW interactions. For the purpose of this study, the heterostructure comprising of BEDT-TTF/C$_{60}$ (ETC$_{60}$) and P3DDT/C$_{60}$ (DTC$_{60}$) complexes (monolayer structure) was built using the Avogadro[48] and Gauss View5 packages[49]. For the ETC$_{60}$ section of the heterostructure, the crystal coordinates were used from the Cambridge Crystallographic Data Centre[50]. The monolayer structures of the ETC$_{60}$ and DTC$_{60}$ complexes were formed by removing the structures of DTC$_{60}$ and ETC$_{60}$, respectively from those of the heterostructure. The heterostructure contains total of 690 atoms out of which 396 belongs to ETC$_{60}$ and remaining 294 associated with DTC$_{60}$, respectively.

**Data availability**. The data that support the findings of this study are available from the online data repository Figshare with identifier https://doi.org/10.6084/m9.figshare.5005331.v1 Remaining data that support the findings are available from the corresponding author upon reasonable request.

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

## Acknowledgements

Work at Temple University (S.R.) was supported by the Army Research Office—Young Investigator Program (W911NF-15-1-0610, material design/self-assembly). We would like to thank Jie Yin from Department of Mechanical Engineering, Temple University for his help on the discussion of strain dependent electrical measurements. H.C., M.L.K., and part of the computational resources were supported as part of the Center for the Computational Design of Functional Layered Materials, an Energy Frontier Research Center funded by the U.S. Department of Energy, Office of Science, Basic Energy Sciences under Award #DE-SC0012575. A portion of the computations is performed on Blue Waters sustained-petascale computing project, which is supported by the National Science Foundation (awards OCI-0725070 and ACI-1238993) and the state of Illinois. This research used resources of the National Energy Research Scientific Computing Center, a DOE Office of Science User Facility supported by the Office of Science of the U.S. Department of Energy under Contract No. DE-AC02-05CH11231. H.C., V.K.Y., and M.L.K. also acknowledge discussions and interactions with Dr. Richard C. Remsing and Dr. Santosh Mogurampelly, Department of Chemistry, Temple University.

## Author contributions

All authors discussed the results and commented on the manuscript. B.X. carried out experiments and wrote the paper. H.C., V.K.Y., and M.L.K. took the simulation. Z.Z. proposed the systhesis method of the film. S.R. designed and guided the project.

## Additional information

**Competing interests:** The authors declare no competing financial interests.

**Change history:** A correction to this article has been published and is linked from the HTML version of this paper.

