## [Peer Review File · Nature Communications]

Reviewers' comments:

Reviewer #1 (Remarks to the Author):

See attached file.

Reviewer #2 (Remarks to the Author):

The authors present a manuscript on 2D organic heterostructures. The topic is very recent and the level of interest as well many question regarding fabrication, manipulation, device layout, or exploratory research regarding diverse properties present a high degree of novelty. The manuscript is well written, however, there are many doubts regarding experimental approaches and their impact on the results discussion. Some questions that arise are:

1) DMF has been used as solvent, but it is very difficult to be removed just by evaporation. Moreover is a solvent that acts as charge traps. How was it removed?

2) It is unclear if the mixture of the two different donors with C60 gives a charge transfer (CT) salt or just a solution with a mixture of the two materials. If a CT salt is formed, it is necessary additional chemical analysis as evidence.

3) The energy band diagram in fig. 1f (and the respective discussion) is not appropriate since it does not corresponds to the layout of the stacking layers.

4) How do authors explain the "disappearance" of part of the absorption band in DTC60 spectra, in the mixture of DTC60 with ETC60? If there are no additional chemical interactions between the compounds, the spectra of the mixture should be the combination of the 2 individual spectra (see for instance J. Am. Chem. Soc. 2015, 137, 7104–7110).

5) The vertical and horizontal measurements are not done in the same device or in a device with the same structure (the vertical device has additional layers). Therefore, the conclusions taken can have substantial different reasons for the different device performance (ex: in the vertical layout was used a cathode and an anode, whereas in the horizontal layout the same electrode was used; this has implications on the charge collection).

6) The authors present in fig. 2f the behavior of samples with light intensity up to 7 mW/cm², showing a linear behavior. However, magnetic measurements were performed under a much more intense light (120 mW/cm²), raising doubts about the linear behavior of samples under that light intensity.

Therefore, I think there are substantial technical and scientific doubts and the manuscript is not sound enough to be published on nature communications.

Reviewer #3 (Remarks to the Author):

In this manuscript, Xu et al. reported the molecular heterostructures based on charge-transfer crystals of P3DDT or BEDT-TTF donor with C60 acceptor. The large area nanosheets were synthesized from solution and then transferred by modified Langmuir-Blodgett method. The interfacial coupling between DTC60 and ETC60 were systematically investigated with external fields including ferroelectric and magnetic, enhancement of current and capacitance was observed. Devices based on such heterostructures also showed very good flexibility, piezoresistance coefficient was calculated. The idea of tunable opto-electronic-magneto coupling is very interesting for the organic heterostructures, and recently the authors published 3 papers (Sci. Adv. 2015, 1, e1501264; Nano Lett. 2016, 16, 2851–2859; Chem. Mater. 2016, 28, 2441–2448) on this topic with CT crystals. For this manuscript, I think more discussion and some necessary improvements are needed before reach the high requirement of “Nature Communications”, detailed comments see below:

1, The crystal structure of nanosheets need more description, we can find the structure of ETC60 from Chem. Mater. 2016, 28, 2441–2448. The simulated packing of DTC60 from Nano Lett. 2016, 16, 2851–2859 is not the layer by layer segregated stacking as shown in Fig 1e. There is not enough evidence for this point, and the SEAD image in figure 1d is not clear also without crystal plane information.

2, For the photoresponse in horizontal and vertical devices, how about the size of these devices? In figure 2 and related discussions, how many devices are fabricated and measured? The obtained on/off ratio and photo-responsivity are average value or not? The authors need to clarify the repeatability of devices, as the transport properties of organic crystals are highly depends on the quality of sample and sometimes varied obviously from each other.

3, The molecular heterostructure here is much more complicated (each nanosheet is consist by stacking of two components) and different from the commonly reported inorganic atomic crystals (Nature Reviews Materials 2016, 1, 16042). The description of interface here needs more supporting since the whole paper is focus on the interfacial coupling. There are more donor/acceptor interface (with charge transfer) inside the nanosheets as they are typically 8 and 23 nm (actually there are almost 10 monolayers), and only one interface between nanosheets,

which kind of interface is dominate in the opto-electronic-magneto properties? The interface in the heterostructure here is contact as C60 to C60, or P3DDT to BEDT-TTF? I think the dipoles might be different and results in different ferroelectric field effect. Thus, control experiments based on individual DTC60, ETC60 or even pure P3DDT, BEDT-TTF, C60 samples are necessary to distinguish the origin of MC effect and etc.

PS. The page numbers for Ref. 6 and Ref 13 are lost.

Reviewer 1 Comments

The authors have used the solvent vapor evaporation assisted by LB approach to make a heterojunction of two charge transfer layers. The author measure the optoelectronic properties along the horizontal and vertical orientations, and measure the magnetic field, ferroelectric field and mechanical effects on the devices. However, most of the conclusions and analyses are not supported with convincing experiments. There are several critical issues in the article that need to be clarified. So I do not suggest to publish this article in Nature Communication.

1. According to the AFM picture (Figure 1b), the thickness of the two layers is 8nm and 23nm, there is no other experiment to tell us if the thickness is controllable or not.
2. According to Figure 1.e, the molecular chains for BEDT-TTF and P3DDT are along the horizontal orientation and C₆₀ is between the layers, there should be some support experiment to prove these structures.
3. The band alignment of Figure 1.f is not enough for readers to understand the electron transfer characteristic in the devices. And the author have not given us the entire device band alignment to show the carriers transport characteristics. There should be a band alignment of the whole device rather than the individual band alignments of the materials.
4. According to the Figure 2.a, we can see that the DTC60 get a photoresponse around 300-600nm rather than the near-infrared area (800-1000nm). So I speculate that the point that author pointed out that the DTC60 nanosheets' spectrum absorption extends from UV to the near-infrared region is unconvincing.
5. According to Figure 2.b, it seems that the horizontal and vertical arrangements are two devices, there is not data about the thickness of the two devices and we are not sure that the different properties of photoresponse result from the different orientations. Also we can see that in the horizontal model, the electrons transfer in the same materials and in the vertical one the electrons transferred through different layers, and the channel length in both devices are different. Therefore the current should be different. So these differences of on/off ratios cannot be the evidence of the anisotropic photoresponse properties of the materials.
6. It does indicate that whether Figure 3.b is in the dark or under light illumination.
7. In Figure 3.c, the line color in the inset is not same as the big one. For example, in the main figure there is a red line, but we cannot see it in the inset. Besides, there is no explanation about why the capacitance is affected by the magnetic field at high frequency, but not so obvious at low frequency.
8. In Figure 3.e, it is said that there is no light dependent tendency at high frequency due to the relaxation of dipoles, there should be some support experiment to prove this explanation.
9. According to Figure 4.a, actually there is no obviously difference between the device with or without P(VDF-TrFE). The unit step of the current is different in the two pictures in Figure 4.a, Maybe it should change to use the same unit step. Also we can see that the on current is unstable

Point to point response

Response to Reviewer #1:

Reviewer #1 (Remarks to the Author):

The authors have used the solvent vapor evaporation assisted by LB approach to make a heterojunction of two charge transfer layers. The author measure the optoelectronic properties along the horizontal and vertical orientations, and measure the magnetic field, ferroelectric field and mechanical effects on the devices. However, most of the conclusions and analyses are not supported with convincing experiments. There are several critical issues in the article that need to be clarified.

1. According to the AFM picture (Figure 1b), the thickness of the two layers is 8nm and 23nm, there is no other experiment to tell us if the thickness is controllable or not.

Response:

Thank you very much for your kind comments.

1) As shown in the following Fig. 1, the formation of high quality nanosheets is dictated by the following factors: the large surface tension of water molecules, water-miscible and spreading ability of dimethylformamide (DMF) solvent, the selective ratio of DMF/water, the solvent for the organic materials with relatively low vapor pressure.

Fig. 1 Growth scheme of the nanosheet.

2) The mixed solvent of DMF/water is of key importance for the homogeneously spreading of molecular ETC₆₀ solution (1,2-dichlorobenzene solvent) on the surface of water (Fig. 2). According to Marangoni flow, when a drop of solution is dropped onto the surface of high-surface-energy phase, local surface tension gradients generate around the boundary of the dropped solution, leading to the surficial flow of the solution towards the higher surface tension part. [1,2] The tendency of the formation of organic film on the surface of water is determined by the spreading pressure S , which is the difference between the work of adhesion $W_{1,2}$ between phases and the work of cohesion $W_{2,2}$ of the phase under consideration.

$$S = W_{1,2} - W_{2,2}$$

Equally, the spreading pressure can be expressed as the difference of the surface tensions along the three-phase contact line, that is the surface tensions σ_1 and σ_2 of the dropped solution and water, and the interfacial tension $\sigma_{1,2}$ of the two phase:

$$S = \sigma_1 - \sigma_2 - \sigma_{1,2}$$

For 1,2-dichlorobenzene (1,2-DCB) on the water surface, the spreading pressure is $-4.25 \text{ dyne cm}^{-1}$. [3] Thus, wetting is not complete, but forming drops. By contrast, as DMF can

be miscible in both water and 1,2-DCB, the mixed water/DMF solution can spread 1,2-DCB solution into film.

Fig. 2 Spreading pressure dependent film formation mechanism.

Fig. 3 Photo images of nanosheet on water (a) and 50 v%DMF/50 v%water surface.

3) Moreover, 1,2-DCB has a very low vapor pressure leading to much lower evaporation rate enabling the homogeneously nucleation. By controlling the ratio of DMF and water, uniform crystallized two-dimensional film can be formed (Fig. 4, and Fig. 5).

Fig. 4 Morphology and composition of ETC₆₀ nanosheet. (a-b) OM images. The scale bar

is 100 μm . (c-f) SEM images and EDS of carbon and sulfide. (g) Atomic ratio of carbon and sulfide of the film on Si substrate.

Fig. 5 Morphology and composition of DTC₆₀ nanosheet. (a-b) OM and SEM images. The scale bar in Fig. 5a is 100 μm . (c-d) TEM images and SAED pattern.

4) By controlling the concentration and volume of organic solution, the thickness of the film can be controlled (Fig. 6). The thickness increases from 8 to 40 nm with the increase of the concentration of DTC₆₀ from 5 to 21 mg/mL at a volume of 20 μL . The further increase of the volume to 100 μL at the concentration of 21 mg/mL leads to the thickness of \sim 200 nm.

Fig. 6 Concentration and volume dependent thickness change of DTC₆₀ nanosheet.

Fig. 7 Concentration dependent thickness change of DTC₆₀ nanosheet.

2. According to Figure 1.e, the molecular chains for BEDT-TTF and P3DDT are along the horizontal orientation and C₆₀ is between the layers, there should be some support experiment to prove these structures.

Response:

Thank you for your valuable comment.

1) Our synchrotron data reveals the segregated stacking of BEDT-TTF and C₆₀, confirming the layered structure as shown in the following Fig. 8. The crystal structure of (BEDT-TTF)₂-C₆₀ was resolved with the space group C2/c and the unit cell are $a = 25.87 \text{ \AA}$, $b = 21.87 \text{ \AA}$, $c = 9.91 \text{ \AA}$, and $\alpha = 90^\circ$, $\beta = 105.13^\circ$, $\gamma = 90^\circ$ with the asymmetric unit cell contains half C₆₀ molecule and one BEDT-TTF molecule. The detailed structure refinement results are summarized in Table 1. The BEDT-TTF and C₆₀ molecules are packed with a stoichiometric ratio of 2:1 and the C₆₀ molecule is wrapped up by a pair of BEDT-TTF molecules (Fig. 8b). In addition, one C₆₀ molecule is contacted with six neighboring BEDT-TTF molecules (Fig. 8b), and two adjacent C₆₀ molecules with distances of S(BEDT-TTF)⋯C(C₆₀): 3.43–3.49 Å, C(BEDT-TTF)⋯C(C₆₀): 3.17–3.39 Å,

and C(C₆₀)...C(C₆₀): 3.28 Å, which is less than the sum of van de Waals radius of the contact atoms, thus leading to significant intermolecular interaction. The interaction between an n-orbital of sulfur atoms from the BEDT-TTF molecules and a π*-orbital of C₆₀ molecules is suggested to be responsible for the charge transfer interaction. Detailed short intermolecular contacts between C₆₀ and BEDT-TTF are shown in Table 2.

Fig. 8 Crystal stacking structure of BEDT-TTF and C₆₀.

Table 1. Data collection and refinement of (BEDT-TTF)₂C₆₀ complex.

Crystal data		
C ₆₀ ·2(C ₁₀ H ₈ S ₈)		$F(000) = 3008$
$M_r = 1489.89$		$D_x = 1.828 \text{ Mg m}^{-3}$
Monoclinic, $C2/c$		Synchrotron radiation, $\lambda = 0.800 \text{ \AA}$
$a = 25.871 (5) \text{ \AA}$		Cell parameters from 4529 reflections
$b = 21.868 (4) \text{ \AA}$		$\theta = 0.8\text{--}28.1^\circ$
$c = 9.910 (2) \text{ \AA}$		$\mu = 0.96 \text{ mm}^{-1}$
$\beta = 105.13 (3)^\circ$		$T = 100 \text{ K}$
$V = 5412 (2) \text{ \AA}^3$		Rod, black
$Z = 4$		$0.2 \times 0.02 \times 0.02 \text{ mm}$
Data collection		
MAR300 diffractometer	CCD	4529 independent reflections
Radiation source: SER-CAT 22BM, APS, ANL, USA		4498 reflections with $I > 2\sigma(I)$
ω scans		$\theta_{\max} = 28.7^\circ$, $\theta_{\min} = 1.4^\circ$
Absorption correction: multi-scan SCALEPACK (Otwinowski et al., 2003)		$h = 0\text{--}31$
$T_{\min} = 0.977$, $T_{\max} = 0.981$		$k = 0\text{--}26$

4529 measured reflections	$l = -11 \rightarrow 11$
Refinement	
Refinement on F^2	180 restraints
Least-squares matrix: full	Hydrogen site location: inferred from neighbouring sites
$R[F^2 > 2\sigma(F^2)] = 0.035$	H-atom parameters constrained
$wR(F^2) = 0.089$	$w = 1/[\sigma^2(F_o^2) + (0.0409P)^2 + 22.9395P]$ where $P = (F_o^2 + 2F_c^2)/3$
$S = 1.04$	$(\Delta/\sigma)_{\max} = 0.001$
4529 reflections	$\Delta)_{\max} = 0.61 \text{ e } \text{\AA}^{-3}$
350 parameters	$\Delta)_{\min} = -0.36 \text{ e } \text{\AA}^{-3}$

Table 2 Short intermolecular contacts between C₆₀ and BEDT-TTF.

Atom1	Atom2	Length(Å)	Length-VdW(Å)
C14_1*	C8_1	3.284	-0.116
C13_1	S5_2*	3.473	-0.027
C17_1	S5_2	3.481	-0.019
C26_1	S5_2	3.428	-0.072
C27_1	C37_2	3.191	-0.209
C27_1	C38_2	3.169	-0.231
C6_1	C33_2	3.393	-0.007
C15_1	C35_2	3.188	-0.212
C20_1	S6_2	3.494	-0.006
S3_2	S8_2	3.555	-0.045

*_1: C60; *_2: BEDT-TTF

2) In fact, the crystalline structures of polythiophene (P3DDT)-C₆₀ have already be reported in our previous paper. The P3DDT self-organizes through the π - π interactions and stacks in-planar structure as follows. The donor P3DDT and the acceptor C₆₀ are alternatively stacking forming the segregated structure. [4-6]

Fig. 9 Layered stacking structure of P3DDT- C₆₀.

3) The XRD pattern (the following Fig. 10) demonstrates that P3DDT (300) crystalline face of DTC₆₀ are along the horizontal direction with P3DDT molecular chain in the in-plane orientation. A peak at ~21° from ETC₆₀ crystal appears, demonstrating that molecular chain of BEDT-TTF is in the in-plane orientation which is consistent with the structure we mentioned in the main text. In addition, the horizontal stacking of the molecular chains is the most stable free-standing structure on water surface due to the side-chain interaction on the water surface. The electrical transport data also support this stacking structure.

Fig. 10 XRD pattern of ETC₆₀, DTC₆₀ and ETC₆₀-DTC₆₀ nanosheet.

3. The band alignment of Figure 1f is not enough for readers to understand the electron transfer characteristic in the devices. And the author have not given us the entire device band alignment to show the carriers transport characteristics. There should be a band

alignment of the whole device rather than the individual band alignments of the materials.

Response :

Thank you for your valuable suggestion. In charge transfer (CT) system, the CT complex has new energy band different from the donor and acceptor component. The donor HOMO has a larger contribution to the HOMO of CT, the CT LUMO correlates more to the LUMO of the acceptor.[7] We have obtained the HOMO and LUMO of the formed heterostructure by first principle calculation. The electronic density of states (DOS) and the partial density of states (PDOS) of the ETC₆₀-DTC₆₀ are shown in the following Fig. 11a-2c, as are the same quantities for the BEDT-TTF/C₆₀ (ETC₆₀), and P3DDT/C₆₀ (DTC₆₀) alone, in the same configuration as in the heterostructure. The energies are shifted by the respective Fermi energy. The formation of the ETC₆₀ and the DTC₆₀ heterostructure results in shifting of the LUMO level closer to the Fermi energy. This decreases the HOMO-LUMO gap, and thus, contributes to the charge transfer in the system. HOMO is comprised mainly of the 2p states of S atoms in BEDT-TTF molecules of ETC₆₀, while the 2p states of C atoms in C₆₀ molecules of DTC₆₀ dominate the LUMO band (see Fig. 11b).

Fig. 11 a, Electronic density of states (DOS) plots. Solid curve denotes the DOS of ETC₆₀/DTC₆₀ heterostructure (black), red curve for ETC₆₀, complex and blue curve for DTC₆₀ complex. The energies have been shifted with respect to their Fermi energies. b-c, Projected density of states (PDOS) plots. Solid curves denotes the PDOS of ETC₆₀/DTC₆₀ heterostructure for S (red). In Fig. 11b, the dotted curve is for C (blue) of BEDT-TTF, and C₆₀ of the DTC₆₀ complex respectively. In Fig. 11c, the dotted curve is for C (blue) of ETC₆₀ complex. The energies have been shifted with respect to their Fermi energies.

4. According to the Figure 2a, we can see that the DTC₆₀ get a photoresponse around 300-600nm rather than the near-infrared area (800-1000nm). So I speculate that the point that author pointed out that the DTC₆₀ nanosheets' spectrum absorption extends from UV to the near infrared region is unconvincing.

Response :

Thank you for your kind comment! According to the absorption spectra, the photoresponse could cover from UV to visible region (near 700 nm). We have made relevant revision in the manuscript.

5. According to Figure 2.b, it seems that the horizontal and vertical arrangements are two devices, there is not data about the thickness of the two devices and we are not sure that the different properties of photoresponse result from the different orientations. Also we can see that in the horizontal model, the electrons transfer in the same materials and in the vertical one the electrons transferred through different layers, and the channel length in both devices are different. Therefore the current should be different. So these differences of on/off ratios cannot be the evidence of the anisotropic photoresponse properties of the materials.

Response:

Thank you for your suggestion.

1) The two devices were fabricated from the same batch of nanosheets. The thickness data is shown in Figs. 1b and 1c of the main text that the thickness of ETC₆₀ and DTC₆₀ nanosheets is 23 and 8 nm, respectively.

2) For the horizontal orientation, the electrodes are deposited parallel and at the ends of both nanosheets. For the vertical orientation, the electrodes are deposited at the top and bottom of the nanosheets. The properties are taken along different orientation as shown for the large difference (over 10³) of photocurrent response along the horizontal and vertical orientations in Fig. 2d and Fig. 2f of the main text. The measurement schematic figure is shown as following Fig. 12:

Fig. 12 Scheme of devices for vertical orientation (left figure) and horizontal orientation (right figure).

To exclude the influence of the structures for horizontal and vertical directions on the optoelectronic properties, we measured the conducting-AFM on the same device of the same structure to verify the anisotropic conductivity as shown in Fig. 13. The conductivity for horizontal orientation is much larger than that of the vertical orientation, which is consistent with the discussion in our main text. In Fig. 11b, as the electrode is in the left, thus, the left part of Fig. 11b shows high conductivity than that of the right part. However, for Fig. 11d, as the electrode is at the bottom, the film shows homogeneous conductivity.

Fig. 13 Conducting-AFM of the same device measured for the horizontal (a, b) and vertical (c, d) orientations. a and c, AFM topography image. b and d, current image. The scale bar is 1 μm . The loaded voltage is 0.5 V.

3) From the same material for both horizontal and vertical directions, the charge transfer interactions along the vertical direction dominates the anisotropic characteristics. The same electrode materials for both horizontal and vertical direction, and therefore they are electron-only devices, which exhibit the anisotropic properties. [8]

4) The difference of channel length has been considered. In consideration of the dimensions, the calculated resistivity for the horizontal and vertical orientations is 9.2×10^7 and $2.5 \times 10^{11} \Omega \text{ cm}$, respectively. As the photoresponsibility doesn't depend on channel length, different photoresponsibility along the two orientations verifies the existence of anisotropic photoresponse in this study.

5) Here, the anisotropic properties are based not on materials but on devices. All the discussion are based on the same device. For both the horizontal and vertical orientations, the charges transport through all the BEDT-TTF, P3DDT and C_{60} layer. The results come from the average effect of the combination of BEDT-TTF, P3DDT and C_{60} molecules. We have made relevant revision in the main text.

6. *It does indicate that whether Figure 3.b is in the dark or under light illumination.*

Response:

Thank you for reminding us! The results of Fig. 3b is measured under dark. We have made relevant revision in the main text.

7. *In Figure 3.c, the line color in the inset is not same as the big one. For example, in the main figure there is a red line, but we cannot see it in the inset. Besides, there is no explanation about why the capacitance is affected by the magnetic field at high frequency, but not so obvious at low frequency.*

Response:

1) The color representation is not right in the previous version. The color of the line in the main figure of Fig. 3c is red, and the inset is brown. We have corrected it in the revised main text.

2) The capacitance can be influenced by magnetic field at low frequency as shown in the

main figure of Fig. 3c and the following figure (Fig. 14) under the frequency of 100-10000 Hz. Here, we show capacitance change under high frequency in order to compare with that of inorganic materials.

Fig. 14 Frequency dependent capacitance change under different magnetic field

8. In Figure 3.e, it is said that there is no light dependent tendency at high frequency due to the relaxation of dipoles, there should be some support experiment to prove this explanation.

Response:

Thank you for your suggestions!

1) With the increase of frequency, dielectric properties go through four polarization processes:

$1 \sim 10^4$ Hz, interfacial and space charge polarization;

$10^4 \sim 10^{10}$ Hz, dipolar polarization;

$10^{10} \sim 10^{14}$ Hz, ionic polarization;

$10^{14} \sim 10^{18}$ Hz, electronic polarization.

As our measurement is between $1 \sim 10^7$ Hz, we will discuss the first two polarization processes. Fig. 15 shows the space charge polarization and dipolar polarization process under ac electric field.

Fig. 15 Schematic representation of different mechanism of polarization. a) Space charge polarization. b) Dipolar polarization.

At low frequency, a number of conduction mechanisms (different species of charge carriers and different carrier mobilities) for the charges to accumulate at interfaces correspond with charges moving towards opposite direction forming interfacial or space charge polarization (Fig. 15a). At high frequency, the charge cannot follow the switch of ac electric field. Only dipolar can be polarized. However, as shown in Fig. 15b, there is only a limited rotation of dipolar side groups leading to low dielectric constant.

2) For our charge transfer materials, as shown for Fig. 3e in the main text, capacitance decreases with the increase of frequency. It can be illustrated as shown in the following Fig. 16. In the long-range ordered crystallized nanostructures, charge transfer across the interface induce the generation of dipoles (triplet exciton). The measurement of capacitance can quantify the extent of the alignment of dipole moment. It is essentially a measure of the dipole moment. [9] At low frequency of electric field, the macroscopic dipoles (the charge polarization) can be aligned along the direction of electric field, inducing large capacitance. So, under low frequency, light induced increased charge transfer could convert to dipoles with enhanced capacitance. With the increase of frequency at a low range, the alignment of macroscopic dipoles (the charge polarization) can still follow the ac field. However, at high frequency, the dipoles can't follow the ac electric field, as a consequence, the dipoles remain randomly oriented. Thus, the capacitance decrease, inducing dielectric relaxation.[10] So under high frequency, although light could induce the increase of charge transfer. But the random dipoles can't contribute to the change of capacitance.

Fig. 16 (a) the dipole alignment along the electric field for the device. (b) The status of dipole at low frequency. (c) The status of dipole at high frequency.

9. According to Figure 4.a, actually there is no obviously difference between the device with or without P(VDF-TrFE). The unit step of the current is different in the two pictures in Figure 4.a, Maybe it should change to use the same unit step. Also we can see that the on current is unstable for the device without P(VDF-TrFE). So longer time is needed to test the real properties in the device.

Response:

We have prepared new devices and made the measurement for a longer time as shown in the following Fig. 17. The unstable current is due to the change of magnetic field. Nevertheless, according to magnetoconductance formula $MC = [j(B) - j(0)]/j(B)$, $j(B)$ and $j(0)$ are current density with and without magnetic field. The change tendency of MC for device with P(VDF-TrFE) is larger. We have made relevant revision in main text.

Fig. 17 Current change with the magnetic field off/on for the device with and without P(VDF-TrFE) underneath ETC₆₀ and DTC₆₀ nanosheets.

10. In Figure 4.b, the line is not fitted well with the dot and the line then does not have clear meaning. Also there is a new structure that the P(VDF-TrFE) layer is inserted between the two layers. No explanation is given about this structure and there is no other experiment to test this structure.

Response:

- 1) We have plotted the fitted line in Fig. 4b. The MC increases with the increase of P(VDF-TrFE) layer thickness till the thickness of 8 nm.
- 2) Two different structure of device are compared to reveal the mechanism of interface coupling. One structure from bottom to top is P(VDF-TrFE) layer, ETC₆₀ nanosheet and DTC₆₀ nanosheet, respectively. The other structure from bottom to top is ETC₆₀ nanosheet, P(VDF-TrFE) layer and DTC₆₀ nanosheet, respectively. When P(VDF-TrFE) layer is between ETC₆₀ nanosheet and DTC₆₀ nanosheet, it prevents the coupling between ETC₆₀ nanosheet and DTC₆₀ nanosheet with smaller MC.
- 3) The structure scheme for P(VDF-TrFE) layer between ETC₆₀ nanosheet and DTC₆₀ nanosheet is shown as follows:

Fig. 18 Schematic figure for P(VDF-TrFE) layer between ETC₆₀ nanosheet and DTC₆₀ nanosheet.

The use of different stacking sequence of P(VDF-TrFE) layers (one is underneath the nanosheets, the other is between the nanosheets) contains two purposes:

One is to investigate the influence of the ferroelectric field of P(VDF-TrFE) layer on the charge transfer of the nanosheets. The ferroelectric field from the P(VDF-TrFE) layer enhances the CT density and MC effect (Fig. 4a-4b in the main text). As shown in Fig. 4c of the main text, dielectric constant for the heterostructures with polarized P(VDF-TrFE) is much higher than that of heterostructures without P(VDF-TrFE). Ferroelectric field effect from P(VDF-TrFE) layer can enhance the CT and the alignment of dipoles in ETC₆₀ nanosheets

The other is to investigate the interfacial coupling behavior between ETC₆₀ nanosheet and DTC₆₀ nanosheet. The device with P(VDF-TrFE) layer between ETC₆₀ and DTC₆₀ nanosheets is weaker than that with P(VDF-TrFE) layer underneath the nanosheets due to the decreased interfacial coupling between ETC₆₀ nanosheet and DTC₆₀ nanosheet by the inserting of P(VDF-TrFE) layer in the middle layer. This is also verified by the higher resistance for the device with P(VDF-TrFE) layer in the middle than that underneath the nanosheets.

We have plot relevant data in the supporting information (Figs. S16 and S17)

11. The article says that the loaded strain is far below its limit to avoid the destruction of the device, but there is no experiment about the exact limit data about the device or the materials.

Response:

We make this assesment from the stress-strain relationship in Fig. 5a that the stress-strain always keeps linear relationship. It is predicted and verified by experiments that two-dimensional film could own large strain ability of over 10%.[4] Moreover, our previous results on P(VDF-TrFE)-ETC₆₀ two dimensional nanosheet shows a strain of 30% (the following Fig. 19). The strain could also induce resistance change (the following Fig. 20). Here, in order to avoid the fracture of the nanosheets and keep the mechanical properties reversible, we haven't loaded large strain. We have deleted that sentence in the manuscript.

Fig. 19 Strain-stress relationship of P(VDF-TrFE) (a), P(VDF-TrFE)-ETC₆₀ film (b).

Fig. 20 Current-voltage curve of P(VDF-TrFE)-ETC₆₀ film under tension strain.

12. There should be some experimental result to tell us the exact relationship about the tension and the performance of the device, such as the linear relationship rather than just tell us with the tension increase, the current decrease.

Response:

Thank you for your valuable suggestion. In fact, we have already shown the tension-conductivity relationship in Fig. 5d, which is not linear. There is two process. For the first process, the conductivity decreases slowly as the strain reaches 4.3% at a stress of 0.085 Mpa. For the first part, as the stress increases further, the conductivity decreases quickly. The piezoresistance coefficient is $-4.4 \times 10^{-6} \text{ Pa}^{-1}$ below the strain of 4.3%, and it decreases to $-4.1 \times 10^{-6} \text{ Pa}^{-1}$ under higher strain.

References:

1. Poulard, C. *et al. Europhys. Lett.* **80**, 64001 (2007).
2. Morita, T. *et al. Appl. Phys. Express* **2**, 111502 (2009).
3. Demond, A. H. *et al. Environ. Sci. Technol.* **27**, 2318 (1993).
4. Xu, B. *et al. Sci. Adv.* **1**, e1501264 (2015).
5. Xu, B. *et al. Nano Lett.* **16**, 2851-2859 (2016).
6. Xu, B. *et al. Chem. Mater.* **28**, 2441-2448 (2016).
7. Akinwande, D. *et al. Nature Commun.* **5**, 5678 (2014).
8. Ren, S. *et al. Nano Lett.* **11**, 3998 (2011).
9. Senanayak, S. P. *et al. Phys. Rev. B* **85**, 115311 (2012).
10. Kulkarni, C. *et al. J. Am. Chem. Soc.* **137**, 3924 (2015).

Reviewer #2 (Remarks to the Author):

The authors present a manuscript on 2D organic heterostructures. The topic is very recent and the level of interest as well many question regarding fabrication, manipulation, device layout, or exploratory research regarding diverse properties present a high degree of novelty. The manuscript is well written, however, there are many doubts regarding experimental approaches and their impact on the results discussion. Some questions that arise are:

1. DMF has been used as solvent, but it is very difficult to be removed just by evaporation. Moreover is a solvent that acts as charge traps. How was it removed?

Response:

Thank you for your kind comment!

1) As shown for the growth scheme in the following Fig. 1, DMF is mixed with water as substrate for the lifting of ETC₆₀ and DTC₆₀ nanosheet by the large surface tension of water and superior dispersion ability of DMF. It's not used as a solvent to dissolve ETC₆₀ and DTC₆₀. So the possibility of DMF entering the crystal lattice of ETC₆₀ and DTC₆₀ is very small.

Fig. 1 Growth scheme of the LB membrane method.

2) The boiling point of DMF is 153 °C. If the annealing temperature is too high, the organic materials will be degraded. Thus, a moderate annealing temperature is needed. So, here, the nanosheet was annealed at 60°C for 4 h and then 120 °C for 20 min to evaporate the solvent under vacuum. It is critical to remove the solvent to decrease the influence of solvent on the properties of the nanosheet. Our synchrotron data also confirms that the solvent is removed from the lattice as shown in the following Fig. 2.

Fig. 2 Crystal stacking structure of BEDT-TTF and C₆₀.

3) The solvent charge trapping will have great influence on the device performance for field-effect transistors where gate voltage is loaded. However, here, our device are all based on two electrodes. The influence of solvent charge trapping is beyond our discussion. This is an interesting challenge for further understanding of the electrical behavior. Nevertheless, we will make further investigation later.

4) The contact between the nanosheet and DMF/water mixed solution is monolayer. Moreover, all the organic donor and acceptor can't be dissolved in DMF. If DMF enters the nanosheet lattice, the content should be very small. The two-dimensional compression during the formation of film can also expel DMF into water [1].

5) If DMF forms charge trap in the nanosheet, large current hysteresis should be observed. However, as shown in the following Fig. 3, only limited hysteresis originating from the intrinsic properties of the nanosheets can be observed.

Fig. 3 Hysteresis current-voltage curve for vertical device.

2. *It is unclear if the mixture of the two different donors with C₆₀ gives a charge transfer (CT) salt or just a solution with a mixture of the two materials. If a CT salt is formed, it is necessary additional chemical analysis as evidence.*

Response:

Thank you for your valuable suggestion.

1) In this work, as shown in the above Fig.1, ETC₆₀ nanosheet was prepared first. Then, DTC₆₀ nanosheet was transferred from the surface of water onto ETC₆₀ nanosheet. They were prepared layer by layer. The formation of CT salt is confirmed indirectly by the existence of magnetoconductance (MC), where CT forms between donor and acceptor, facilitating the magnetic field induced current change by tuning the intersystem crossing from singlet CT to triplet CT.

2) XRD diffraction pattern as shown in the following Fig. 4 confirms the formation of charge transfer salts. In addition, the formation of CT salt can also be confirmed by the SAED pattern in the inset of Fig. 1d where crystalline diffraction arrays appear.

Fig. 4 XRD diffraction patterns of ETC₆₀, DTC₆₀ and ETC₆₀-DTC₆₀ nanosheets.

3) Raman spectra of C₆₀, ETC₆₀, DTC₆₀ and ETC₆₀-DTC₆₀ nanosheet are shown in the following Fig. 5. Symmetric (Ag) mode of C₆₀ at 1465 cm⁻¹ is used to test the ionicity of the charge transfer complex. The shift of Ag peak and the broadening of the peak for charge transfer complex compared to single C₆₀ nanosheet confirm the formation of charge transfer crystals.

Fig. 5 Raman spectra of C_{60} , ETC_{60} , DTC_{60} and ETC_{60} - DTC_{60} nanosheet excited by 532 nm laser.

3. The energy band diagram in fig. 1f (and the respective discussion) is not appropriate since it does not corresponds to the layout of the stacking layers.

Response:

Thank you for your kind suggestion. We have made relevant revision in the main text. In charge transfer (CT) system, the CT complex has new energy band different from the donor and acceptor component. The donor HOMO has a larger contribution to the HOMO of CT, the CT LUMO correlates more to the LUMO of the acceptor.[2]

Fig. 6 a, Electronic density of states (DOS) plots. Solid curve denotes the DOS of ETC_{60}/DTC_{60} heterostructure (black), red curve for ETC_{60} , complex and blue curve for DTC_{60} complex. The energies have been shifted with respect to their Fermi energies. b-c, Projected density of states (PDOS) plots. Solid curves denotes the PDOS of ETC_{60}/DTC_{60} heterostructure for S (red). In Fig. 6b, the dotted curve is for C (blue) of BEDT-TTF, and C_{60} of the DTC_{60} complex respectively. In Fig. 6c, the dotted curve is

for C (blue) of ETC₆₀ complex. The energies have been shifted with respect to their Fermi energies.

4. How do authors explain the "disappearance" of part of the absorption band in DTC₆₀ spectra, in the mixture of DTC₆₀ with ETC₆₀? If there are no additional chemical interactions between the compounds, the spectra of the mixture should be the combination of the 2 individual spectra (see for instance *J. Am. Chem. Soc.* 2015, 137, 7104–7110).

Response:

1) As shown in Fig. 4, the absorption at 465 nm of the mixture is a combination of the absorption of the two individual spectra.

Fig. 4 Absorption spectra of ETC₆₀, DTC₆₀ and ETC₆₀-DTC₆₀ in 1,2-DCB solvent.

2) Absorption of ETC₆₀, DTC₆₀ and ETC₆₀-DTC₆₀ nanosheets are shown in the following Fig. 5. The DTC₆₀ and ETC₆₀-DTC₆₀ nanosheets show strong absorption bands at around 465~500 nm, and the absorption intensity of ETC₆₀-DTC₆₀ nanosheets for this absorption band is an overlap of both ETC₆₀ and DTC₆₀ nanosheets.

Fig. 5 Absorption spectra of ETC₆₀, DTC₆₀ and ETC₆₀-DTC₆₀ nanosheets.

5. The vertical and horizontal measurements are not done in the same device or in a device with the same structure (the vertical device has additional layers). Therefore, the

conclusions taken can have substantial different reasons for the different device performance (ex: in the vertical layout was used a cathode and an anode, whereas in the horizontal layout the same electrode was used; this has implications on the charge collection).

Response:

1) The two devices were fabricated from the same batch of nanosheets. The thickness data is shown in Figs. 1b and 1c of the main text that the thickness of ETC₆₀ and DTC₆₀ nanosheets is 23 and 8 nm, respectively.

2) For the horizontal orientation, the electrodes are deposited parallel and at the ends of both nanosheets. For the vertical orientation, the electrodes are deposited at the top and bottom of the nanosheets. The properties are taken along different orientation as shown for the large difference (over 10³) of photocurrent response along the horizontal and vertical orientations in Fig. 2d and Fig. 2f of the main text. The measurement schematic figure is shown as following Fig. 6:

Fig. 6 Scheme of devices for vertical orientation (left figure) and horizontal orientation (right figure).

3) To exclude the influence of the structures and electrodes for horizontal and vertical directions on the optoelectronic properties, we measured the conducting-AFM on the same device of the same structure to verify the anisotropic conductivity as shown in the following Fig. 7. The conductivity for horizontal orientation is much larger than that of the vertical orientation, which is consistent with the discussion in our main text. In Fig. 7b, as the electrode is in the left, thus, the left part of Fig. 7b shows high conductivity than that of the right part. However, for Fig. 7d, as the electrode is at the bottom, the film shows homogeneous conductivity.

Fig. 7 Conducting-AFM of the same device measured for the horizontal (a, b) and vertical (c, d) orientations. a and c, AFM topography image. b and d, current image. The scale bar is 1 μm. The loaded voltage is 0.5 V.

6. The authors present in fig. 2f the behavior of samples with light intensity up to 7 mW/cm², showing a linear behavior. However, magnetic measurements were performed under a much more intense light (120 mw/cm²), raising doubts about the linear behavior of samples under that light intensity. Therefore, I think there are substantial technical and scientific doubts and the manuscript is not sound enough to be published on nature communications.

Response:

1) The measurements in Fig. 2f and Fig. 3f are photocurrent response and capacitance, respectively. The photoresponse is due to light induced generation of charge carriers (electrons and holes) while the capacitance change is due to light induced triplet excitons (longer lifetime).

2) For photoresponse, the measurement is often taken under a low light intensity to avoid the light-soaking effect as shown in Fig. S5e.[4] We have measured the photocurrent up to the light intensity of 100 mW/cm² for both vertical and horizontal orientations as shown in the following Fig. 8. As discussed in Fig. 2f of the main text, the light intensity dependent photoresponse current includes two parts. At low light intensity lower than 7 mW/cm² and at high light intensity of higher than 7 mW/cm², the photoresponse current and power law dependence follows linearly behavior $I \propto P^\theta$ with θ close to 1, indicating there is few trapped photogenerated charge carriers and the recombination of charge carriers is dominated by monomolecular recombination at low light intensity.[5, 6]

Fig. 8 Two parts of photoresponse current as a function of light intensity for both vertical and horizontal orientations. The change of linear behavior happens at ~7 mW/cm².

However, as a whole measurement, the relationship is not linear (in the following Fig. 9) due to the loss of charge carriers via bimolecular recombination and space charge limited photocurrent from the unbalanced transport of electrons and holes.[7,8]

Fig. 9 Photoresponse current as a function of light intensity for both vertical and horizontal orientations.

3) As shown in Fig. 3f of the main text, the capacitance shows large change under a small light intensity of 10 mw/cm^2 and the relation to light intensity is also not linear behavior.

Reference:

- [1] Guennouni, Z. *et al.* *Langmuir* 32, 1971 (2016).
- [2] Salmerón-Valverde, A. *et al.* *CrystEngComm* 17, 6227 (2015).
- [3] Xu, B. *et al.* *Chem. Mater.* **28**, 2441 (2016).
- [4] Jacobs-Gedrim, B. *et al.*, *ACS Nano* 8, 514 (2014)
- [5] Kind, H., *et al.*, *Adv. Mater.* **14**, 158 (2002).
- [6] Koster, L. J. A. *et al.*, *Appl. Phys. Lett.* **87**, 203502 (2005).
- [7] Koster, L. J. A. *et al.*, *Appl. Phys. Lett.* **88**, 052104 (2006).
- [8] Mihailetchi, V. D. *et al.*, *Phys. Rev. Lett.* **94**, 126602 (2005).

Reviewer #3 (Remarks to the Author):

In this manuscript, Xu et al. reported the molecular heterostructures based on charge-transfer crystals of P3DDT or BEDT-TTF donor with C₆₀ acceptor. The large area nanosheets were synthesized from solution and then transferred by modified Langmuir-Blodgett method. The interfacial coupling between DTC₆₀ and ETC₆₀ were systematically investigated with external fields including ferroelectric and magnetic, enhancement of current and capacitance was observed. Devices based on such heterostructures also showed very good flexibility, piezoresistance coefficient was calculated. The idea of tunable opto-electronic-magneto coupling is very interesting for the organic heterostructures, and recently the authors published 3 papers (Sci. Adv. 2015, 1, e1501264; Nano Lett. 2016, 16, 2851–2859; Chem. Mater. 2016, 28, 2441–2448) on this topic with CT crystals. For this manuscript, I think more discussion and some necessary improvements are needed before reach the high requirement of “Nature Communications”, detailed comments see below: 1, The crystal structure of nanosheets need more description, we can find the structure of ETC₆₀ from Chem. Mater. 2016, 28, 2441–2448. The simulated packing of DTC₆₀ from Nano Lett. 2016, 16, 2851–2859 is not the layer by layer segregated stacking as shown in Fig 1e. There is not enough evidence for this point, and the SEAD image in figure 1d is not clear also without crystal plane information.

Response:

Thank you for your kind suggestion.

- 1) Due to the large structure difference, both the donor P3DDT, BEDT-TTF and the acceptor C₆₀ are alternatively stacking forming segregated structure. [1-3] For (BEDT-TTF)C₆₀, BEDT-TTF and C₆₀ stack layer by layer. But for P3DDT and C₆₀, they stack separately. We have made relevant revision in the main text.
- 2) The following Fig. 1 shows the XRD pattern of ETC₆₀, DTC₆₀ and ETC₆₀-DTC₆₀ nanosheets. The observed P3DDT (300) crystalline face of DTC₆₀ indicates that (300) crystalline face is in-plane crystalline face, thus molecular chain of P3DDT is orientated along the horizontal orientation. A peak at ~21° from ETC₆₀ crystal appears, demonstrating that (600) crystalline face of ETC₆₀ crystal is in-plane crystalline face, thus the molecular chain of BEDT-TTF is along the horizontal orientation which is consistent with the structure we mentioned in the main text.

Fig. 1 XRD pattern of ETC₆₀, DTC₆₀ and ETC₆₀-DTC₆₀ nanosheets.

3) The TEM and SAED images for single ETC₆₀ and DTC₆₀ layers are shown in the following Fig.2, demonstrating the crystallized structure.

Fig. 2 TEM and SAED pattern of ETC₆₀ (a) and DTC₆₀ (b) layers.

4) Our calculation based on first-principles density functional theory (DFT) have revealed the stacking structure of the nanosheet heterostructure (the following Fig. 3). Each ETC₆₀ nanosheet crystalline unit contains 6 molecules of BEDT-TTF and 4 molecules of C₆₀, while that of DTC₆₀ contains 2 molecules of P3DDT and 2 molecules of C₆₀. The orthorhombic crystalline unit cell of ETC₆₀/DTC₆₀ has lattice parameters of $a=26.71 \text{ \AA}$, $\alpha=90^\circ$, $b=32.51 \text{ \AA}$, $\beta=90^\circ$, $c=58.90 \text{ \AA}$, $\gamma=90^\circ$.

Fig. 3 Stacking structure of nanosheet heterostructure along different orientation. z axis is the out-of-plane orientation.

2, For the photoresponse in horizontal and vertical devices, how about the size of these devices? In figure 2 and related discussions, how many devices are fabricated and measured? The obtained on/off ratio and photo-responsivity are average value or not? The authors need to clarify the repeatability of devices, as the transport properties of organic crystals are highly depends on the quality of sample and sometimes varied obviously from each other.

Response:

Thank you for your valuable comment!

1) The vertical electrode area is 0.01 cm². The distance of the horizontal electrodes range from 50 to 150 μm.

2) We have made over 160 devices to check the photoresponse behavior. In fact, the response behavior is very repeatable. The data in Fig. 2 of the main text is the averaged value. We have added more data as follows (Fig. 4 in the following).

Fig. 4 Two parts of photoresponse current as a function of light intensity for both vertical and horizontal orientations. The change of linear behavior happens at ~7 mW/cm².

Here, in our experiment, P3DDT and C₆₀ form two dimensional crystals, they can withstand the oxygen and moisture atmosphere under ambient conditions. The performance for the device is more stable than ordinary organic P3DDT-C₆₀ solar cells.

3, *The molecular heterostructure here is much more complicated (each nanosheet is consist by stacking of two components) and different from the commonly reported inorganic atomic crystals (Nature Reviews Materials 2016, 1, 16042). The description of interface here needs more supporting since the whole paper is focus on the interfacial coupling. There are more donor/acceptor interface (with charge transfer) inside the nanosheets as they are typically 8 and 23 nm (actually there are almost 10 monolayers), and only one interface between nanosheets, which kind of interface is dominate in the opto-electronic-magneto properties? The interface in the heterostructure here is contact as C₆₀ to C₆₀, or P3DDT to BEDT-TTF? I think the dipoles might be different and results in different ferroelectric field effect. Thus, control experiments based on individual DTC₆₀, ETC₆₀ or even pure P3DDT, BEDT-TTF, C₆₀ samples are necessary to distinguish the origin of MC effect and etc.*

Response :

Thank you for your valuable suggestion! We have made relevant revision in the main text and the supporting information.

1) Magnetoconductance (MC) is the influence of conductance by external magnetic field. Under small magnetic field, MC can only happens in organic charge tranfer systems or p-n junctions [4-5], where the intermolecular charge transfer interaction is much weaker than that of the intramolecular charge transfer (exciton). For pure P3DDT, BEDT-TTF and C₆₀, no MC can be observed under small magnetic field.

2) Here, the energy matching of ETC₆₀ and DTC₆₀ nanosheets makes the charge transport much easier than those of the single layer nanosheet or the nanosheet of pure donor or acceptor as demonstrated in the following:

a) If the upper layer DTC₆₀ is changed to pure P3DDT layer, the photoreponse will be greatly decreased due to the mismatch of energy level as shown in the Fig. 1f of the main text and the following Fig. 5.

Fig. 5 a, Optical microscope of P3DDT/ETC₆₀ nanosheets. The scale bar is 100 μ m. b, Photoresponse of P3DDT/ETC₆₀ nanosheets.

b) The resistivity of ETC₆₀-DTC₆₀ along the horizontal and vertical orientations is 9.2×10^7 and $2.5 \times 10^{11} \Omega \text{ cm}$, respectively. As shown in the following Fig. 6, the resistivity of single layer ETC₆₀ is over 100 times larger, revealing the enhancement of charge transport by energy matching of the heterostructure.

Fig. 6 Resistivity along the vertical (a) and horizontal (b) orientations of single ETC₆₀ nanosheet.

3) In fact, both individual DTC₆₀ and ETC₆₀ nanosheets show MC effect. The DTC₆₀ nanosheets have negative MC under dark condition, in which the current decreases with the loading of external magnetic field as shown in the following Fig. 7. However, the ETC₆₀ nanosheet has positive MC under dark as shown in the following Fig. 8. From the influence of P(VDF-TrFE) layer on the MC effect, we can find that when P(VDF-TrFE) layer is underneath both ETC₆₀-DTC₆₀ layers, the device shows higher MC due to the large coupling between ETC₆₀-DTC₆₀ layers. However, when P(VDF-TrFE) layer is inserted between ETC₆₀-DTC₆₀ layers, the MC is much smaller, revealing that the interface between ETC₆₀-DTC₆₀ layers also play a key role for the coupling here. Although both the individual ETC₆₀ and DTC₆₀ nanosheets show MC, the introduced heterostructure enhance the matching of energy level with much higher charge transport and photoresponse behavior as discussed above.

Fig. 7 MC of DTC₆₀ nanosheet. a, Current change with magnetic field ON and OFF. b, Magnetic field dependent MC change.

Fig. 8 MC of ETC₆₀ nanosheet. a, Current change with magnetic field ON and OFF. b, Magnetic field dependent MC change.

4) As discussed in XRD pattern of question 1, the molecular chain of P3DDT and BEDT-TTF are both aligned along the horizontal orientation. The large crystal lattice difference between P3DDT and BEDT-TTF makes it difficult to accommodate the two molecules in the interface. From both the XRD pattern and related properties characterization, the growth of DTC₆₀ segregated stacking structure epitaxial from C₆₀ layer of ETC₆₀ will be more energy favorable (or stable) than from BEDT-TTF layer. In combination with the stacking structure as shown in the above Fig. 3, we can conclude that the structural interface between the heterostructure is C₆₀ layer of ETC₆₀ and P3DDT layer of DTC₆₀.

5) In order to understand the nature of CT in the heterostructure (ETC₆₀-DTC₆₀), we performed first principles calculations. The electronic density of states (DOS) and the partial density of states (PDOS) of the ETC₆₀-DTC₆₀ are shown in the following Figs. 9, as are the same quantities for the BEDT-TTF/C₆₀ (ETC₆₀), and P3DDT/C₆₀ (DTC₆₀) alone, in the same configuration as in the heterostructure. The energies are shifted by the respective Fermi energy (the following Figs. 9a-9c). The formation of the ETC₆₀ and the DTC₆₀ heterostructure results in shifting of the LUMO level closer to the Fermi energy. This decreases the HOMO-LUMO gap, and thus, contributes to the charge transfer in the system. HOMO is comprised mainly of the 2p states of S atoms in BEDT-TTF molecules of ETC₆₀, while the 2p states of C atoms in C₆₀ molecules of DTC₆₀ dominate the LUMO band (the following Figs. 9b). This suggests that charge transfer is occurring between the n-orbitals of the S atoms in BEDT-TTF of ETC₆₀ and the π^* orbitals of C atoms in C₆₀ of DTC₆₀. Thus, the CT state is responsible for inter-complex charge transfer (between the ETC₆₀ and DTC₆₀ complexes) in the heterostructure. There is also a significant contribution of the intra-complex charge transfer within the ETC₆₀, but with a higher HOMO LUMO gap than the inter-complex CT state (the following Figs. 9c). Other CT pathways of lower probability may also exist (the following Figs. 10-11). The PDOS shows significant density of BEDT-TTF C atom 2p states in the HOMO, supporting the possibility of a π to π^* transition between BEDT-TTF and C₆₀ of ETC₆₀ and DTC₆₀ respectively. Intermolecular charge transfer in this manner results in spatial separation of charge in the CT state, with accumulation of holes on the BEDT-TTF molecules and electrons on the C₆₀ molecules of DTC₆₀. This is evidenced by the charge density isosurfaces of the HOMO and LUMO states in Fig 9d.

Fig. 9 a, Electronic density of states (DOS) plots. Solid curve denotes the DOS of ETC₆₀/DTC₆₀ heterostructure (black), red curve for ETC₆₀, complex and blue curve for DTC₆₀ complex. The energies have been shifted with respect to their Fermi energies. b-c, Projected density of states (PDOS) plots. Solid curves denotes the PDOS of

ETC₆₀/DTC₆₀ heterostructure for S (red). In Fig. 9b, the dotted curve is for C (blue) of BEDT-TTF, and C₆₀ of the DTC₆₀ complex respectively. In Fig. 9c, the dotted curve is for C (blue) of ETC₆₀ complex. The energies have been shifted with respect to their Fermi energies. d, Charge density isosurface of the HOMO (red) and LUMO (blue) bands of the ETC₆₀/DTC₆₀ heterostructure.

Fig. 10 Projected density of states (PDOS) plots. In Fig. 10a, the solid curves denotes the PDOS of ETC₆₀/DTC₆₀ heterostructure for S (red), and dotted curve for C (blue) of the DTC₆₀ complex respectively. In Fig. 10b, the solid curves denotes the PDOS of ETC₆₀/DTC₆₀ heterostructure for C (red), and dotted curve for C (blue) of DTC₆₀ and ETC₆₀ complex respectively. The energies have been shifted with respect to their Fermi energies.

Fig. 11 Projected density of states (PDOS) plots. Solid curves denotes the PDOS of ETC₆₀/DTC₆₀ heterostructure for S (red) and dotted curve for C (blue) of the DTC₆₀ and ETC₆₀ complex respectively. The energies have been shifted with respect to their Fermi energies.

4. PS. The page numbers for Ref. 6 and Ref 13 are lost.

Response :

Thank you for your valuable suggestion! We have added the page numbers already.

Reference:

1. Xu, B. *et al. Sci. Adv.* **1**, e1501264 (2015).
2. Xu, B. *et al. Nano Lett.* **16**, 2851-2859 (2016).
3. Xu, B. *et al. Chem. Mater.* **28**, 2441-2448 (2016).
4. Hu, B. *et al. Adv. Mater.* **21**, 1500 (2009).
5. Yang, D. *et al. Adv. Funct. Mater.* **23**, 2918 (2013).

Reviewers' Comments:

Reviewer #1 (Remarks to the Author):

Manuscript No. NCOMMS-16-21023A

Thanks the authors to address the some concerns about the manuscript.

Unfortunately I find several major missing points. So I CANNOT support to be considered for publication in Nature Comm.

1. The key issue of the manuscript is the CT at the interface of (BEDT-TTF)/C60:(P3DDT)/C60 nanosheets.

As requested before, the energy diagram is ultimately required to clarify the interface properties. PDOS is not a convincing technique to reveal the interface nature, due to its inadequacy in dealing with the van der Waals interactions via DFT. Experimentally XPS/UPS and other techniques are necessary to interrogate the issue. More relevant information can be found in M. Oehzelt, et al. Science Advances 1, e1501127 (2015)

2. The substrate effect is completely missed.

Substrate has a huge impact on the formation of individual nanosheets and heterostructures. Unfortunately there is no any discussion on the topic. To minimize the substrate impact, self-assembled monolayers are usually introduced.

Several relevant papers are listed below.

1. D. Q. Liu, et al., Ange. Chem. 62, 6222 (2013)
2. K. Chen, et al., J. Phys. Chem. C. 116, 8259 (2012)
3. X. M. Wang, et al., Adv. Mat. 23, 2464 (2011)

Reviewer #2 (Remarks to the Author):

The authors have made changes in the manuscript and cleared some doubts and it is now ready to be published by nature communications.

Reviewer #3 (Remarks to the Author):

The authors have addressed the reviewer's concern, I suggest the acceptance of the manuscript.

Point to point response**Response to Reviewer #1:**

Reviewer #1 (Remarks to the Author):

1. The key issue of the manuscript is the CT at the interface of (BEDT-TTF)/C60:(P3DDT)/C60 nanosheets. As requested before, the energy diagram is ultimately required to clarify the interface properties. PDOS is not a convincing technique to reveal the interface nature, due to its inadequacy in dealing with the van der Waals interactions via DFT. Experimentally XPS/UPS and other techniques are necessary to interrogate the issue. More relevant information can be found in M. Oehzelt, et al. Science Advances 1, e1501127 (2015)

Response: Thank you very much for your valuable suggestion to improve the manuscript!

1) We have carried out cyclic voltammetry (CV) experiment in a three-electrode cell to clarify the energy diagram of ETC₆₀ and DTC₆₀ nanosheets. Here, the glassy carbon electrode is used as the working electrode, Pt electrode is used as counter electrode, and Ag/AgCl electrode in saturated KCl water solution is used as the reference electrode. All the electrodes are cleaned and blown dry by nitrogen gas before use. The 0.1 M tetrabutylammonium tetrafluoroborate (Sigma Aldrich) is used as the electrolyte and dissolved in dehydrated acetonitrile (ACN). The stability of Ag/AgCl electrode is checked by ferrocene (Sigma Aldrich) internal standard substance. The ETC₆₀ and DTC₆₀ solution is self-assembled onto the working electrode and dried at 40 °C in glove box to form uniformly coated nanosheets. As shown in the following Fig. 1, ferrocene (Fc) shows stable oxidation peak at 0.54 V. While two resolvable oxidation peaks with onsets

at 0.34 and 0.90 V are observed for ETC₆₀ and DTC₆₀ nanosheets. Thus, we can obtain the HOMO level of ETC₆₀ and DTC₆₀ nanosheets according to the equation:

$$E_{\text{HOMO}} = [-\exp(E_{\text{onset}}(\text{vs. Ag/AgCl}) - E_{\text{onset}}(\text{Fc/Fc}^+ \text{ vs. Ag/AgCl}))] - 4.8\text{eV}$$

Figure 1, The CV curve of ferrocene.

Figure 2, The CV curve of ETC₆₀ and DTC₆₀ nanosheets.

By considering the bandgap from the absorption and photoluminescence spectra, we can get the HOMO and LUMO of ETC₆₀ and DTC₆₀ nanosheets as shown in the following Fig. 3a. The alignment of HOMO and LUMO of ETC₆₀ and DTC₆₀ nanosheets facilitates the charge transfer between the two nanosheets as discussed in the manuscript.

Figure 3, Band alignment (a) and charge transfer (b) between of ETC₆₀ and DTC₆₀ nanosheets.

2) While considering the DFT calculations may underestimate the band gaps [1-2], here we have utilized DFT to calculate the band alignment of single ETC₆₀ and DTC₆₀ nanosheet for revealing the nature of charge transfer interaction within the heterostructures (ETC₆₀-DTC₆₀). The computed results are in qualitative agreement with the experimental ones. Here, DFT can accurately predict the following aspects, which have only been elaborated in the main text:

- a) Reduction in the band gap of the heterostructure of ETC₆₀ and DTC₆₀ complexes as compared to the individual components.
- b) Charge transfer occurring within the complexes, (from S atom of ETC₆₀ to C atom of DTC₆₀) and is a dominant effect as compared to the charge transfer in individual complexes (ETC₆₀ and DTC₆₀).

In summary, the computed results by the DFT method are in qualitative agreement with the experiments. Both experiment results of CV curves and DFT calculation confirm the charge transfer interaction between ETC₆₀ and DTC₆₀.

2. The substrate effect is completely missed. Substrate has a huge impact on the formation of individual nanosheets and heterostructures. Unfortunately there is no any discussion on the topic. To minimize the substrate impact, self-assembled monolayers are usually introduced. Several relevant papers are listed below.

*1. D. Q. Liu, et al., *Ange. Chem.* 62, 6222 (2013)*

*2. K. Chen, et al., *J. Phys. Chem. C.* 116, 8259 (2012)*

*3. X. M. Wang, et al., *Adv. Mat.* 23, 2464 (2011)*

Response:

Thank you very much for your kind comments! As a follow-up and clarification to this comment, we did not use the solid substrates during the formation of nanosheets, instead the nanosheets are created at the liquid interface.

For the conventional chemical vapor deposition (CVD) method, the substrate is necessary and indeed essential. However, the substrate has large impact on the self-assembly of nanosheets, especially two-dimensional films. The choice of lattice-matched substrate is a necessary which limits the scalable preparation of two-dimensional films [3-5]. Moreover, the weak interlayer interaction of two-dimensional films made by CVD leads to the island growth rather than continuous monolayers [6]. Due to these drawbacks, here, we have developed a modified Langmuir-Blodgett (LB) method to facilitate the layer-by-layer self-assembly of scalable freestanding films on the surface of water/dimethylformamide (DMF) mixed solution.

As shown in Fig. 1a of the main text and the following Figure 4, ETC₆₀ and DTC₆₀ in 1,2-dichlorobenzene (DCB) solvent is spread on the surface of water/DMF mixed

solution with the sequential self-assembly into nanosheets. Thus, the effect of substrate can be neglected, endowing the nanosheet with superior performance here.

Figure 4, The growth scheme of the nanosheet.

References:

- [1] Perdew, J. P. (1985), Density functional theory and the band gap problem. *Int. J. Quantum Chem.*, 28: 497–523. doi:10.1002/qua.560280846
- [2] Pribram-Jones, A., Gross, D. A. & Burke, K. *Annu. Rev. Phys. Chem.* **66**, 283-304 (2015).
- [3] Zang, Y., Zhang, F., Di, C-a. & Zhu, D. *Mater. Horiz.* **2**, 140-156 (2015).
- [4]. Huang, X. & Zhang, H. *Sci. China Mater.* **58**, 5-8 (2015).
- [5]. Lee, C-H. *et al. Adv. Mater.* **26**, 2812-2817 (2014).
- [6]. Geim, A. K. & Grigorieva, I. V. *Nature* **499**, 419-425 (2013).

Reviewer #2 (Remarks to the Author):

The authors have made changes in the manuscript and cleared some doubts and it is now ready to be published by nature communications.

Response:

Thank you very much!

Reviewer #3 (Remarks to the Author):

The authors have addressed the reviewer's concern, I suggest the acceptance of the manuscript.

Response:

Thank you very much!

Reviewer #1 (Remarks to the Author):

Although the authors mainly replied my questions with some modified method for the energy diagram, I still feel that the overall quality is still below the norm of NC, particularly in terms of device performance, i.e., responsivity and response time. They are far inferior to those of TMDCs based materials.

Point to point response

Comments:

Manuscript No. NCOMMS-16-21023B

Response to Reviewer #1 (Remarks to the Author):

1. Although the authors mainly replied my questions with some modified method for the energy diagram, I still feel that the overall quality is still below the norm of NC, particularly in terms of device performance, i.e., responsivity and response time. They are far inferior to those of TMDCs based materials.

Response: Thank you for your valuable suggestion to improve the quality of our manuscript!

1) By optimizing the device structure and the thickness of each layer, the performance of 2D heterostructure devices is expecting to be improved. However, our submitted manuscript focuses on the anisotropic interfacial coupling and charge-transfer (CT) induced external stimuli (ferroelectric field, magnetic field, optical excitation and mechanical strain) tunable conductance and capacitance of all-organic 2D heterostructures. Some of these performance presented in this submission has not been observed in transition metal dichalcogenide (TMDCs) based materials.

2) Conventional TMDCs based 2D heterostructures are composed of two layers of opposite carrier type. However, in this submission, it is important to investigate the coupling behavior between two pairs of different CT layers, which holds promise to show unique coupling behavior distinct from traditional inorganic counterparts due to

the inter-conversion between singlet and triplet CT states under external stimuli combined with the internal hyperfine interaction and spin-orbit coupling of organic/molecular semiconductors. Moreover, the combination of the distinct CT induced properties from different organic layers, shown in this submission, is not only complementary but also enhances the performance of the whole all-organic system. As demonstrated in our manuscript, two-dimensional organic CT molecular heterostructures can open a new paradigm of materials with tunable opto-electric-magneto-mechanical coupling properties for flexible all-organic electronic/magnetic device applications.

3) For the preparation of conventional TMDCs based materials, the choice of lattice-matched substrate is a necessary which limits the scalable preparation of two-dimensional films [1-3]. Moreover, the weak interlayer interaction of TMDCs made by CVD leads to the island growth rather than continuous monolayers [4]. Although there is some report of the preparation of TMDCs from liquid/liquid interface, they are based on the exfoliated layers from the bulk materials [5]. In comparison, organic CT molecular materials presented here show unique advantages of self-assembly into large-scale ordered superstructures due to the presentation of intermolecular interactions, such as charge-transfer induced electrostatic interaction, van der Waals forces and π - π stacking [6]. Here, we propose a new method - modified Langmuir-Blodgett - to grow molecular heterostructures directly, to facilitate the layer-by-layer self-assembly of scalable freestanding films on the surface of solution. Thus, the effect of substrate can be neglected endowing the large-scale fabrication of

nanosheet here. Our bottom-up growth strategy involves both polymers and small molecules, while it is universal and can be applied for other organic materials.

4) As shown in the following Fig. 1, the BEDT-TTF and C_{60} molecules are alternatively stacked along the horizontal orientation (x and y axis), forming the segregated stacking 2D ETC₆₀ nanosheets with the distance between each layer close to 6.46 Å. The P3DDT layers self-organize through the π - π interactions. The distinct structure of P3DDT and C_{60} molecules lead to the segregated stacking of P3DDT and C_{60} in the 2D DTC₆₀ nanosheets, where the closest distance between the backbone of P3DDT and C_{60} molecules is 8.08 Å. Each layer is composed of only one kind of molecule. By modifying the structure and properties of each layer of molecules, and by molecular-scale engineering of the interface, multiple approaches can be applied to tune the properties of the materials. Moreover, there is not only interface between ETC₆₀ and DTC₆₀ layers, but also intra-layer interface between BEDT-TTF and C_{60} molecules, and P3DDT and C_{60} molecules. The charge transfer of all the interface can be tuned affording lots of freedom to control the properties of the materials.

Fig. 1 Stacking structure of ETC₆₀ nanosheets and DTC₆₀ nanosheets along different axis.

5) Our DFT simulation demonstrates for the first time that there is strong electronic band hybridization and dominant charge transfer interaction between ETC60 and DTC60 layers. As shown in the Fig. 2, the HOMO of two-dimensional molecular heterostructure is comprised mainly of the 2p states of S atoms in BEDT-TTF molecules of ETC60, while the 2p states of C atoms in C60 molecules of DTC60 dominate the LUMO band (Fig. 2a). The charge transfer occurs between the n-orbitals of the S atoms in BEDT-TTF of ETC60 and the π^* orbitals of C atoms in C₆₀ of DTC60.

Fig. 2 a, Projected density of states (PDOS) plots. Solid curves denotes the PDOS of ETC₆₀/DTC₆₀ heterostructure for S (red). The blue curve is for C from C₆₀ of the DTC₆₀ complex respectively. b, Charge density isosurface of the HOMO (red) and LUMO (blue) bands of the ETC₆₀/DTC₆₀ heterostructure.

References

- [1] Zang, Y., Zhang, F., Di, C-a. & Zhu, D. *Mater. Horiz.* **2**, 140-156 (2015).
- [2] Huang, X. & Zhang, H. *Sci. China Mater.* **58**, 5-8 (2015).
- [3] Lee, C-H. *et al. Adv. Mater.* **26**, 2812-2817 (2014).
- [4] Geim, A. K. & Grigorieva, I. V. *Nature* **499**, 419-425 (2013).
- [5] Yu, X., Prevot, M. S., Guijarro, N. & Sivula, K. *Nat. Commun.* **6**, 7596 (2015).
- [6] Xu, B. *et al. Sci. Adv.* **1**, e1501264 (2015).